

# Domain wall problem in the quantum XXZ chain and semiclassical behavior close to the isotropic point

Grégoire Misguich[1,2]⋆, Nicolas Pavloff[3] and Vincent Pasquier[1]

**1** Institut de Physique Théorique, Université Paris Saclay, CEA,
CNRS UMR 3681, 91191 Gif-sur-Yvette, France
**2** Laboratoire de Physique Théorique et Modélisation,
CNRS UMR 8089, Université de Cergy-Pontoise, 95302 Cergy-Pontoise, France
**3** Laboratoire de Physique Théorique et Modèles Statistiques,
CNRS UMR 8626, Univ. Paris-Sud, Université Paris Saclay, 91405 Orsay, France

⋆ gregoire.misguich@cea.fr

## Abstract

We study the dynamics of a spin-$\frac{1}{2}$ XXZ chain which is initially prepared in a domain-wall state. We compare the results of time-dependent Density Matrix Renormalization Group simulations with those of an effective description in terms of a classical anisotropic Landau-Lifshitz (LL) equation. Numerous quantities are analyzed: magnetization ($x$, $y$ and $z$ components), energy density, energy current, but also some spin-spin correlation functions or entanglement entropy in the quantum chain. Without any adjustable parameter a quantitative agreement is observed between the quantum and the LL problems in the long time limit, when the models are close to the isotropic point. This is explained as a consequence of energy conservation. At the isotropic point the mapping between the LL equation and the nonlinear Schrödinger equation is used to construct a variational solution capturing several aspects of the problem.

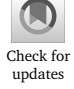

# 1  Introduction

The study of out-of-equilibrium quantum systems constitutes a major research field in condensed matter and quantum statistical physics. The understanding of the dynamics of isolated quantum many-body systems after a quench is one of the important questions in this domain [1, 2]. It addresses the long-time behavior of the system after a sudden change of some external parameter at time $t = 0$, and, for instance, the possible relaxation of local observables to steady values, and the characterization of the associated steady state.

In the field of quantum quenches, the situation of integrable systems is very peculiar [3]. Their dynamics is qualitatively different from that of generic models where the only local conserved quantities are the energy and the momentum. Indeed, in integrable systems the large number of local conserved quantities represents an enhanced memory of the initial state. In situations where a non-integrable system would approach a Gibbs state characterized by a single quantity – the temperature or the energy – an integrable system will keep a much more detailed memory of its initial conditions. When the system is spatially homogeneous, this can give rise to so-called generalized Gibbs ensembles [4]. This also leads to unconventional transport properties (see, e.g., Ref. [5–7]).

In the case of an inhomogeneous quench, the system is initially prepared at $t = 0$ in a

state which is spatially inhomogeneous, while the Hamiltonian driving the unitary evolution at $t > 0$ can be taken to be translation invariant. As an example one may suddenly join at $t = 0$ two macroscopically different and homogeneous states, having, for instance, different particle densities, different magnetizations or different temperatures.

Such protocols are interesting because they can produce current-carrying states, and they allow to address questions about transport (ballistic or diffusive?) or to test hydrodynamic descriptions. In integrable one-dimensional systems, like the spin-$\frac{1}{2}$ XXZ chain or the Bose gas with $\delta$ interactions [8], an important progress has been made recently concerning the dynamics of inhomogeneous states. An hydrodynamic description taking into account the local conserved quantities was developed, which is now called "generalized hydrodynamics" (GHD) [9, 10]. It has lead to many fruitful developments in the last few years [7, 11–14, 14–21], including on the experimental side [22].

One of the applications of GHD is a classic inhomogeneous quench problem [23–25], where an XXZ spin chain is prepared at time $t = 0$ in a domain-wall state where all the spins in the left half of the chain are pointing "up", and the spins in the right half are pointing "down". This setup leads to three regimes, depending on the anisotropy parameter $\Delta$ in the Hamiltonian of the chain. For $|\Delta| < 1$ (easy-plane) the dynamics creates a $z$-magnetization profile which extends ballistically with time. $\Delta = 0$ is a simple limit (Jordan-Wigner mapping to free fermions) where such a ballistic transport can be compared with exact calculations [23]. For $|\Delta| > 1$ (easy-axis) this profile gets frozen at long time [24, 25]. While the GHD can be used to study the system in the ballistic cases in details, it does not predict much more than the absence of ballistic transport for $|\Delta| \geq 1$. Numerical simulations however suggests a diffusive behavior with logarithmic corrections at the isotropic point $\Delta = 1$ [26].

In a recent paper Gamayun *el al.* [27] (we also mention [28]) considered the domain wall problem in a different model, which is a classical ferromagnetic chain in the continuum limit. There, the magnetization $\vec{M}$ is a classical unit vector which depends on time and on a continuous space variable $r$, and its (precession) dynamics is described by the celebrated anisotropic Landau Lifshitz (LL) equation. The initial condition taken in Ref. [27] is a smooth function or $r$ interpolating between $\vec{M} = \vec{e}_z$ at $r = -\infty$ to $\vec{M} = -\vec{e}_z$ at $r = +\infty$, which is the continuum analog of the domain wall in the context of lattice spin chains. The main result of [27] is that the dynamics of $M^z(r, t)$ follows three regimes, and is qualitatively similar to the quantum case: ballistic in the easy-plane regime, frozen in the easy-axis, and diffusive with logarithmic correction in the isotropic model.

In the present paper we present some quantitative comparisons for the domain wall problem i) in the XXZ chain, and ii) in the anisotropic LL system. We investigate numerous quantities: the magnetization ($z$ component but also $x$ and $y$ components), the energy density, the energy current, but also some spin-spin correlation functions or the entanglement entropy in the quantum chain. We analyze the "diffusion" region, of size $\sqrt{t}$ (with multiplicative logarithmic corrections), but also the regime where $r/t$ is finite, and where both the LL and XXZ problems show some nontrivial behavior. These results are obtained using time-dependent density-matrix renormalization group (tDMRG) simulations of the spin-$\frac{1}{2}$ model, as well as numerical and analytical calculations for the LL problem (hydrodynamic approximation, perturbative expansion, or mapping to the nonlinear Schrödinger equation (NLS) and variational ansatz). As an important result, we observe that the similarities between the quantum spin chain and the LL problem are not only qualitative, but semi-quantitative (or even quantitative) close to the isotropic point, in the long time limit, and without any adjustable parameter. As explained in Sec. 3, we argue that this is a simple consequence of energy conservation. Sec. 4 deals with the easy-plane case. It is shown that the LL problem gives a linear $z$-magnetization profile which is identical to that of XXZ problem in the limit $\Delta \to 1^-$. While the presence of a linear profile in the long time limit was known for these two models, we show that they

have exactly the same slope in the limit $\Delta \to 1^-$, and similar finite-time corrections. In Sec. 5 we analyze the LL problem in the limit where the magnetization is close to $M^z = \pm 1$. This perturbative expansion turns out to correctly describe the regime when $|r|/t$ is large for LL, with small amplitude oscillations of $\vec{M}$ around the $z$ axis. But it also describes some aspects of the XXZ problem, like the tail of the $\langle S^z \rangle$ or energy density profiles. In Sec. 6 we discuss the spatial width of the stationary profile in the easy-axis case, and characterize the scaling of long-lived oscillations around this stationary profile, both in LL and XXZ. Section 7 is devoted to the isotropic models ($\Delta = 1$). We compare the $z$-component of the magnetization profile in both problems, and we confirm the finding of [26] and [27] concerning respectively a logarithmic correction to diffusion in the quantum case and in the classical LL case. We also consider the $x$ and $y$ components of the LL magnetization, which show small amplitude oscillations extending beyond the diffusion scale, up to $r/t$ of order one. While the in-plane magnetization is zero (by symmetry) in the quantum problem, the in-plane magnetization of the LL problem can be quantitatively compared with spin-spin *correlations* in the quantum chain (Sec. 7.3). An heuristic argument concerning the (logarithmic) growth of the entanglement entropy in the spin-1/2 model is also given. In Sec. 7.7 we exploit the mapping between the isotropic LL equation and the NLS equation to construct a variational solution which captures several aspects of the isotropic domain problem: not only the diffusive-like expansion of the $z$ component of the magnetization profile, but also the larger scale behavior of several geometrical quantities like the curvature or the torsion associated to the LL to NLS mapping. Finally, the relevance of the self-similar solutions of the isotropic LL equation to the present domain-wall problems is discussed. The mapping from LL to NLS is recalled in Appendices A and B. An explicit calculation of self-similar solution of the LL equation is given in Appendix C.

## 2 The model: domain wall problem in the XXZ chain

### 2.1 XXZ model

We study the evolution of a quantum XXZ spin-1/2 chain, prepared at time $t = 0$ in a domain wall state $|DW\rangle = |\uparrow\uparrow \cdots \uparrow\uparrow\downarrow\downarrow \cdots \downarrow\downarrow\rangle$ where all the spins in the left half of the system are "up" ($S^z = \frac{1}{2}$) and those in the right half are "down" ($S^z = -\frac{1}{2}$). At time $t > 0$ the wave function then evolves according to

$$|\psi(t)\rangle = \exp\left(-i\hat{H}t\right)|DW\rangle, \tag{1}$$

where the Heisenberg Hamiltonian $\hat{H}$ is that of a XXZ chain of length $L$ with open boundary conditions,

$$\hat{H} = -\sum_r \left( \hat{S}_r^x \hat{S}_{r+1}^x + \hat{S}_r^y \hat{S}_{r+1}^y + \Delta \left[ \hat{S}_r^z \hat{S}_{r+1}^z - \frac{1}{4} \right] \right), \tag{2}$$

and $\Delta$ is the anisotropy parameter.[1] $r \in \mathbb{Z}$ labels the lattice sites, but in the next section $r$ will be treated as a continuum variable.

This problem was first studied by Antal *et al.* [23, 29] in the free fermion case ($\Delta = 0$), where an exact analytical solution for the long-time limit of the magnetization profile was obtained. A few years later the problem with $\Delta \neq 0$ was studied numerically by Gobert *et al.* [24] using the time-dependent density-matrix renormalization group (DMRG). For $|\Delta| < 1$ an exact solution for the long-time limit of the magnetization profile was recently obtained [19]

---

[1]The constant term $-\frac{1}{4}\Delta$ is introduced here to set at zero the energy of a fully polarized state in the $z$ direction. The global minus sign in the definition of $\hat{H}$ is here to simplify the connection with the classical *ferromagnetic* LL description, but it has no influence on the dynamics when starting from $|DW\rangle$ at $t = 0$.

using GHD [9,10]. Despite some recent analytical progresses [30], the precise scaling of the front remains unknown at the Heisenberg point $\Delta = 1$, even though recent large-scale numerics suggest that it shows some diffusive behavior, possibly with a multiplicative logarithmic correction [26].

In the following, the data for the XXZ spin chain are obtained using a tDMRG algorithm [31, 32], implemented using the iTensor library [33], as in Ref. [26]. Technically, we typically use systems of size $L = 800$ sites, Trotter time-step $dt = 0.2$ (matrix-product operator scheme $W^{II}$ [34] at order 4 [35]), a matrix-product state (MPS) truncation parameter equal to $10^{-10}$ or $10^{-11}$ and maximum bond dimension from 1000 to 2500. These state-of-the art simulations are pushed up to $t = 300$. On the LL side, the calculations are performed using the software Maple (pdsolve) with systems size up to $L = 3200$, and space and time discretization steps $dx = dt$ from 0.01 to 0.05. We checked that, at the scale of the plots, the results presented here are essentially free of finite-step of finite size errors.

## 2.2 Anisotropic Landau Lifshitz equation

From a different perspective, one can study an analogous domain-wall problem for a *classical* spin model. To this end, we consider the anisotropic Landau-Lifshitz (LL) equation. It describes the (precession) dynamics of a classical XXZ model in the continuum limit [36,37]:

$$\vec{\Omega}_t = \vec{\Omega} \wedge \left[ \vec{\Omega}_{rr} + \delta \left( \Omega^z \vec{e}_z \right) \right], \tag{3}$$

where $\delta$ is the anisotropy parameter, and $r$ has become a continuous variable.[2] Note that the $z$ component of the equation above can be re-written

$$\Omega_t^z + I_r = 0, \tag{4}$$

where

$$I(r,t) = \Omega^y \Omega_r^x - \Omega^x \Omega_r^y \tag{5}$$

is the current associated to the $z$-component of the magnetization.

Since we wish to make some *quantitative* comparison between the local magnetization $\langle S^z \rangle$ of the XXZ chain with $\Omega^z$, we set the norm of the LL magnetization vector to $|\vec{\Omega}| = 1/2$, and one can easily check that the anisotropy should be set to

$$\delta = 2(\Delta - 1). \tag{6}$$

The above equations of motion can also be obtained from the following classical Hamiltonian:

$$H_{\mathrm{LL}} = \frac{1}{2} \int \left[ \left( \partial_r \vec{\Omega} \right)^2 - \delta \left\{ (\Omega^z)^2 - \frac{1}{4} \right\} \right] dr. \tag{7}$$

The constant term ensures, as in the quantum case, that a uniform state polarized in the $z$ direction has zero energy. $\delta > 0$ corresponds to easy-axis cases, $\delta < 0$ to easy-plane, and $\delta = 0$ to the isotropic model. The isotropic LL model is also often called the classical Heisenberg model.

As proposed in Refs. [27,28], for the LL model a natural counter part of the lattice domain wall state is the following smooth initial condition:

$$\vec{M}(r, t = 0) = 2\vec{\Omega}(r, t = 0) = \begin{pmatrix} 1/\cosh(ar) \\ 0 \\ -\tanh(ar) \end{pmatrix}. \tag{8}$$

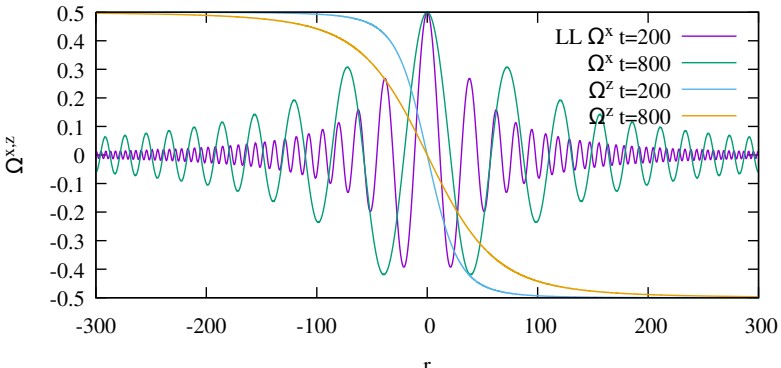

Figure 1: $\Omega^x$ and $\Omega^z$ magnetization profiles for the isotropic LL model ($\Delta = 1$) at $t = 200$ and $t = 800$ (initial condition given in Eq. (8)).

It describes some smooth domain wall which interpolates between $\vec{\Omega} = \frac{1}{2}\vec{e}_z$ at $r \to -\infty$ and $\vec{\Omega} = -\frac{1}{2}\vec{e}_z$ at $r \to +\infty$; $a$ is the inverse width of the initial profile.

For $\delta \neq 0$ it is convenient to define a new space variable $\zeta = r|\delta|^{\frac{1}{2}}$ and a new time variable $\tau = \frac{1}{2}|\delta|\,t$, so that the LL equation becomes:

$$\vec{M}_\tau = \vec{M} \wedge \vec{H}_{\text{eff}}, \quad \text{where} \quad \vec{H}_{\text{eff}} = \vec{M}_{\zeta\zeta} + \text{sign}(\delta)\,M_z\,\vec{e}_z. \tag{9}$$

Now, the inverse width $a$ of the initial conditions is changed to $\tilde{a} = a/\sqrt{|\delta|}$ in the variable $\zeta$. In other words, in the case $|\delta| \ll 1$ we are interested in, we may consider Eq. (9) with an anisotropy parameter fixed to $\pm 1$, but with a very narrow initial profile in terms of $\zeta$ ($\tilde{a} \to \infty$).

## 3 Energy conservation and classical behavior close to $\Delta = 1$

We give here a heuristic argument based on energy conservation, arguing that the behavior of the quantum XXZ chain for $\Delta$ close to 1 gets closer and closer to that of a classical LL problem as the energy and the $\langle S^z \rangle$ profile spread over larger distances.

### 3.1 Energy in the domain wall problem

For the quantum XXZ chain [Eq. (2)], the energy density on one bond can be written as

$$\hat{H}_r = H_r^{\text{iso}} + (\Delta - 1)V_r, \tag{10}$$

with

$$H_r^{\text{iso}} = \frac{1}{4} - \vec{S}_r \cdot \vec{S}_{r+1}$$
$$V_r = \frac{1}{4} - S_r^z S_{r+1}^z.$$

The largest eigenvalue of $\vec{S}_r \cdot \vec{S}_{r+1}$ is $\frac{1}{4}$, corresponding to the triplet states (three-fold degenerate). As for $S_r^z \hat{S}_{r+1}^z$, its largest eigenvalue is also $\frac{1}{4}$, and it corresponds to $|\uparrow\uparrow\rangle$ and $|\downarrow\downarrow\rangle$ (two-fold degenerate). So, the mean values of two terms $H_r^{\text{iso}}$ and $V_r$ appearing in $\hat{H}_r$ are necessarily $\geq 0$. The total mean energy $E_{\text{XXZ}} = \langle\psi(t)|\hat{H}|\psi(t)\rangle$ is conserved along the time

---

[2]We use here the compact notations $\partial_r^2(\Omega^\alpha) \to \Omega_{rr}^\alpha$ and $\partial_t \Omega^\alpha \to \Omega_t^\alpha$ for the space and time derivatives.

evolution and its value is easily computed at $t = 0$, where only the central bond contributes: $E_{XXZ} = \langle DW | \hat{H} | DW \rangle = \frac{1}{2} \Delta$.

We now assume that there exists some length scale $R_\Delta(t)$ over which the energy is spread after time $t$. We further make the simplifying assumption that the energy density is finite and that it varies smoothly with position in this region, and that the energy outside $[-R_\Delta(t), R_\Delta(t)]$ is negligible. The total energy $E_{XXZ}$ can then be written as $E_{XXZ} = 2e(t)R_\Delta(t)$, where

$$e(t) = \langle H^{iso} \rangle + (\Delta - 1) \langle V \rangle \tag{11}$$

is a mean energy density, and $\langle \cdots \rangle$ denotes a quantum average as well as a spatial average over the bonds in the nontrivial region of the system, where the energy is distributed. We conclude that

$$\frac{\Delta}{4R_\Delta(t)} = \langle H^{iso} \rangle + \frac{\delta}{2} \langle V \rangle . \tag{12}$$

Let us examine the implications of this relation when $R_\Delta(t)$ tends to infinity.

- If $\Delta = 1$ ($\delta = 0$), we know [26] that the $\langle S^z \rangle$ profile extends in a diffusive way, with logarithmic corrections. This implies that the energy spreads over a distance which is at least as large as $\mathcal{O}(\sqrt{t})$, and thus $R_\Delta(t) \to \infty$. The energy density therefore tends to zero everywhere when $t \to \infty$. We will see later (in Sec. 7.4) that the energy density in the isotropic spin chain goes to zero as $1/t$, possibly with logarithmic corrections. From Eq. (12) we get that $\langle H^{iso} \rangle$ vanishes when $t \to \infty$. We conclude that, in the nontrivial region $[-R_\Delta(t), R_\Delta(t)]$, the nearest-neighbor correlations asymptotically become that of a ferromagnetic state. The spins thus become locally aligned, but the direction of the magnetization is unconstrained.

- If $\Delta > 1$ ($\delta > 0$) the two terms in the right-hand side of Eq. (12) are positive. If we further assume that $R_\Delta(t)$ is large, then $\langle H^{iso} \rangle$ (as well as $\delta \langle V \rangle$) will be small, of order $\mathcal{O}(R_\Delta(t)^{-1})$. We will see below (Sec. 6) that the $\langle S^z \rangle$ profile extends over a distance of the order of $\delta^{-\frac{1}{2}}$. So, even though $R_\Delta(t)$ may not diverge when $t \to \infty$ at fixed $\Delta > 1$, we have $\lim_{t \to \infty} R_\Delta(t) > \mathcal{O}(\delta^{-\frac{1}{2}})$. So, in the limit of a weak easy-axis anisotropy ($\delta \ll 1$), the state again approaches locally a ferromagnetic state.

- If $\Delta < 1$ we will see that there is a ballistic propagation of the front at velocity $v_{XXZ} = \sqrt{1 - \Delta^2}$, and thus $R_\Delta(t) \simeq t \, v_{XXZ} \to \infty$. So, at long times $\langle H^{iso} \rangle \simeq (1 - \Delta) \langle V \rangle$ and thus $0 \leq \langle H^{iso} \rangle \leq \frac{1}{4} |\delta|$ since $\langle V \rangle \leq \frac{1}{2}$. We see here that when $\Delta$ approaches $1^-$, the nearest-neighbor correlations again become that of a ferromagnetic state.

In all the cases above, where the nearest-neighbor correlation become asymptotically that of a ferromagnetic state, we expect that the spins will become ferromagnetically aligned over *large distances*. These large, almost ferromagnetic, segments of the chain will have a large total spin and should therefore behave semi-classically. This should occur at large times for $\Delta = 1$, but also if we consider the regime where one simultaneously takes the limit of large time and $\Delta \to 1$. In such situations we conjecture that the quantum effects as well as the lattice effects will become weaker and weaker, and that the *classical (anisotropic) LL description should become quantitatively accurate* when compared to the XXZ problem. This conjecture is supported by the data presented below, where we compare the numerical solutions of the (classical) LL equations to numerical results obtained for the (quantum) XXZ spin chain with $\Delta$ close to 1.

Of course the short-time dynamics of the XXZ model is not classical, and we should therefore expect a quantitative agreement between the lattice quantum problem and the classical LL model only for quantities that are independent of the short-distance properties of the initial

domain wall. Another important consequence of the lattice is the existence of a maximum Lieb-Robinson velocity in the quantum chain, whereas arbitrarily high velocities can in principle be observed in the continuum limit. Weak quantum fluctuations will also be present at any finite time, since the energy density is never strictly zero for $t < \infty$. Analyzing in detail the quantum corrections to the classical dynamics is beyond the scope of this paper, but the numerical data presented in this study show that many observables in the XXZ chain behave almost classically at long times.

## 3.2 Initial width of the LL domain wall

Using Eq. (7) one finds that the energy of the initial condition given in Eq. (8) is

$$E_{\text{LL}} = \frac{1}{4}(a + \delta/a). \tag{13}$$

Being conserved by dynamics, the value of this energy will constrain the evolution of the system. In order to be able to make some quantitative comparison between the dynamics of the quantum system and that of the classical system, we chose $a$ such that the classical energy is the same as that of the quantum problem, which is equal to $E_{\text{XXZ}} = \frac{1}{2}\Delta$. In what follows we therefore take $a(\Delta)$ to be solution of $a + 2(\Delta - 1)/a = 2\Delta$. This implies in particular that for $\Delta \to 1$ we have $a \to 2$.

We will successively discuss the three regimes: easy-plane ($0 \leq \Delta < 1$), easy-axis ($\Delta > 1$) and finally the isotropic model ($\Delta = 1$).

# 4 Easy plane regime $\Delta < 1$

It has recently been shown [27] that in the long time limit, the easy-plane LL equation admits solutions of the form $\Omega^z(r, t) = -\frac{r}{2t}|\delta|^{-\frac{1}{2}}$ for $|r| \leq t\sqrt{-\delta}$, and $\Omega^z = \pm 1/2$ outside this interval. Or, equivalently: $M^z(\zeta, \tau) = -\frac{\zeta}{2\tau}$ for $|\zeta/\tau| \leq 2$. We give below a simple derivation of this solution. In Sec. 4.2 we show that, for $\Delta \to 1^-$, the above linear profile in the LL model exactly matches that derived from GHD for the easy-plane XXZ chain. This is the first manifestation of the asymptotic classical behavior conjectured in Sec. 3.1.

## 4.1 Solution of LL – neglecting dispersion effects

When $\vec{M}$ is parametrized using spherical coordinates

$$\vec{M} = \begin{pmatrix} \sin\theta\cos\varphi \\ \sin\theta\sin\varphi \\ \cos\theta \end{pmatrix}, \tag{14}$$

the Eq. (9) becomes [38]

$$\theta_\tau = -2\,\theta_\zeta\,\varphi_\zeta\cos\theta - \varphi_{\zeta\zeta}\sin\theta\,, \tag{15a}$$

$$\varphi_\tau = -\cos\theta\left[\varphi_\zeta^2 + \text{sign}(\delta)\right] + \frac{\theta_{\zeta\zeta}}{\sin\theta}, \tag{15b}$$

where $\text{sign}(\delta) = -1$ in the easy-plane regime, $\text{sign}(\delta) = 1$ in the easy-axis regime and $\text{sign}(\delta) = 0$ for the isotropic Heisenberg case.

The term $\frac{\theta_{\zeta\zeta}}{\sin\theta}$ in (15b) represents dispersive effects. We refer, for instance, to the discussion in [39,40], where the so-called "Riemann problem" for the easy-plane LL equation was studied

in the context of two-component Bose-Einstein condensates. We now make the assumption that it can be neglected in the long time limit. Next, in view of the $\varphi_\zeta^2 - 1$ factor, we look for a solution where $\varphi_\zeta = -1$. This means that the projection of $\vec{\Omega}$ in the $x - y$ plane forms a spiral with a constant pitch, which is perfectly consistent with the data shown in the right panel of Fig. 2.

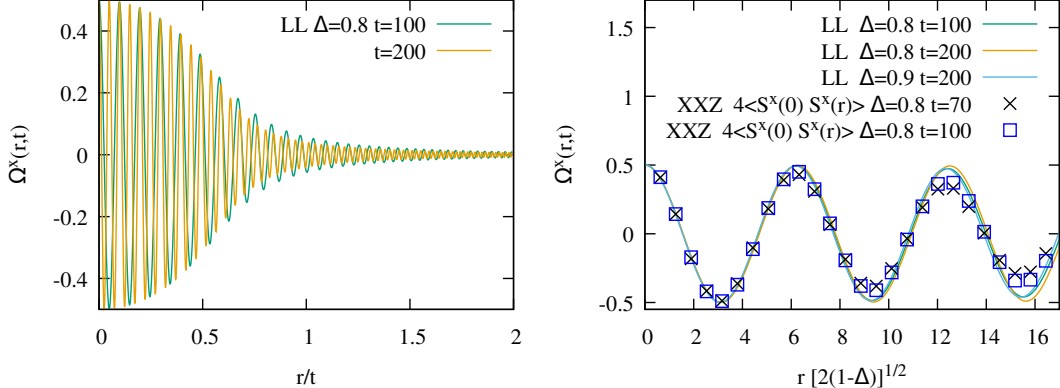

Figure 2: Left panel: $\Omega^x(r, t)$ as a function of the rescaled position $r/t$ for $\Delta = 0.8$ for different times $t = 100$ and $t = 200$. The envelope of the oscillations is approximately the same at $t = 100$ and $t = 200$ (this follows from the fact that $\Omega^z$ scales as $r/t$, see Fig. 3). Right panel: when plotted as a function of $r\sqrt{|\delta|} = r\sqrt{2(1-\Delta)} = \zeta$, the period of the first oscillations turns out to be almost independent of time and $\Delta$, and is numerically close to $2\pi$. This can be understood using the asymptotic solution discussed in Sec. 4.1, which predicts that $|\varphi_\zeta| = 1$. The crosses are spin-spin correlations in the XXZ chain at $\Delta = 0.8$, $t = 70$ and $t = 100$. See Sec. 7.2 for the relation between spin-spin correlations in the quantum chain and the in-plane component of the LL magnetization.

Then Eq. (15b) simplifies to $\varphi_\tau = 0$, and (15a) becomes

$$\theta_\tau = 2\,\theta_\zeta \cos\theta. \tag{16}$$

For $|\zeta/\tau| \le 2$ an obvious solution of the equation above is then

$$\cos\theta = -\frac{\zeta}{2\tau}, \tag{17}$$

which corresponds to $M^z = -\frac{\zeta}{2\tau}$. One can check a posteriori that, in this solution, $\theta_{\zeta\zeta}$ is of order $\mathcal{O}(\tau^{-2})$ and thus becomes negligible at long times. For $|\zeta/\tau| \le 2$, the solution lies at the boundary of the hyperbolicity domain of the easy axis LL system, and is sometimes referred to as a "vacuum region" [41]. Indeed, for this range of values of $\theta$ and $\phi_\zeta$, the model can be mapped onto a system of shallow-water Kaup-Boussinesq equations [40] for which the condition $|\varphi_\zeta| = 1$ corresponds to the absence of the fluid [42]. For $|\zeta/\tau| > 2$ the linear profile makes room for a constant magnetization with $\cos\theta = \pm 1$. Going back to the original $r$ and $t$ variables, this corresponds to a front which propagates at the velocity

$$v_{LL} = \sqrt{-\delta} = \sqrt{2(1-\Delta)}. \tag{18}$$

As can be seen in Fig. 3, the transient time needed to converge to such a profile increases when $\Delta$ approaches $1^-$. Indeed, at $\Delta = 0.99$ and $t = 100$ for instance, the LL profile (as well as the XXZ one) is still relatively far from the asymptotic result. These finite time corrections are the largest in the region corresponding to the $r/t \simeq v_{LL}$ where the linear part of the profile connects to the constant part. Looking at $\Omega^x$ (Fig. 2), the numerical solutions also confirm that $\varphi_\zeta \simeq -1$, as predicted above.

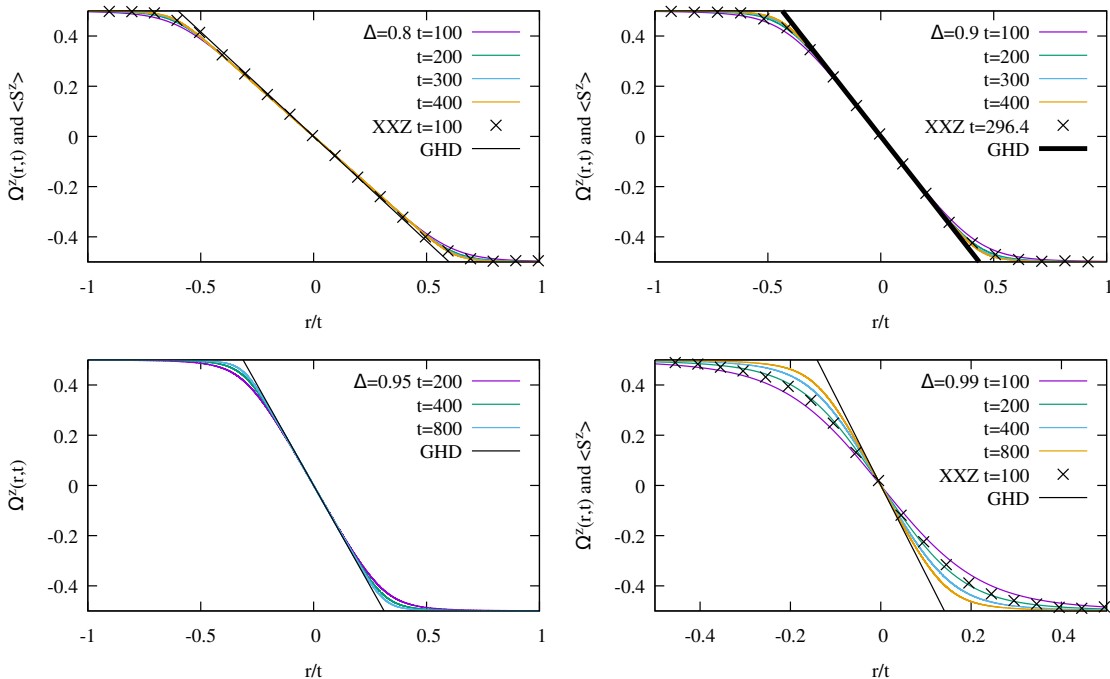

Figure 3: $\Omega^z(r,t)$ from the solution of the classical LL equation, as a function of the rescaled position $r/t$ for $\Delta = 0.8$, $\Delta = 0.9$, $\Delta = 0.95$ and $\Delta = 0.99$ and for different times from $t = 100$ to $t = 800$. The black lines, labeled GHD, represent the long-time limit of the profile for the XXZ chain, $\langle S^z \rangle = -\frac{r}{2t}\left(1 - \Delta^2\right)^{-1/2} = -\frac{r}{2t}(v_{\text{XXZ}})^{-1}$ [19], as predicted by GHD. In the panels associated to $\Delta = 0.8, 9$ and $\Delta = 0.99$, the black crosses represent tDMRG data for the XXZ chain.

## 4.2 Easy plane LL and GHD

For the XXZ spin chain the exact form of the $\langle S^z \rangle$ profile has been computed in the framework of GHD. For a generic values of $\Delta$ [3] it turns out that this profile is a simple linear function of $r/t$ [19]:

$$\langle S^z \rangle = -\frac{r}{2v_{\text{XXZ}}t}. \tag{19}$$

with a velocity given by:

$$v_{\text{XXZ}} = \sqrt{1 - \Delta^2}. \tag{20}$$

So, the LL and XXZ problems both show a linear profile for the $z$ component of the magnetization.

But, in addition, we observe that the slopes of the profiles become *identical* in the limit $\Delta \to 1$, since $v_{\text{XXZ}}/v_{\text{LL}} = \sqrt{(\Delta + 1)/2}$. We argue that this is not an accidental coincidence but a manifestation of the asymptotic classical behavior discussed in Sec. 3.1. Naturally, this match relies on the fact that, on the LL side, the initial configuration has the same energy as the domain wall for the quantum chain.

While the emergence of a linear profile is quite easily understood for the LL model (Sec. 4.1), the fact that such a simple profile emerges from such a complicated model as the XXZ chain (for a generic value of $\Delta$) and from the the GHD equations is quite remarkable. Clearly, the asymptotic equivalence with LL when $\Delta \to 1^-$ – as argued in Sec. 3 – sheds some light on this

---

[3]$\Delta$ may be parametrized as $\Delta = \cos(\Theta)$. We then call *generic* the values of $\Delta$ for which the angle $\Theta$ is *not* of the form $\Theta = \pi Q/P$, with $Q$ and $P$ coprime integers satisfying $1 \leq Q < P$.

question. It would also be interesting to compare further the finite-time behavior of the LL magnetization profile with that of the XXZ chain, in the light of the finite-time corrections to the asymptotic profile close to the edge at $r/t = v_{XXZ}$, as studied in Refs. [19, 43].

# 5 Perturbative expansion

We consider here the classical LL problem and focus on the region where $\vec{M}$ is close to $-\vec{e}_z$. At a given time, this is supposed to hold for sufficiently large $r$. We can therefore write

$$\vec{M} = 2\vec{\Omega}(r,t) = \alpha(r,t)\vec{e}_x + \beta(r,t)\vec{e}_y - \sqrt{1-\alpha^2-\beta^2}\vec{e}_z, \tag{21}$$

with $\alpha$ and $\beta \ll 1$. The equation (3) can be expanded to first order in $\alpha$ and $\beta$. Using the complex variable $z = \alpha + i\beta$, the linearized equation reads

$$2z_t = -iz_{rr} + i\delta z, \tag{22}$$

and admits the following plane wave solutions

$$z(r,t) \;=\; z_0 \exp(i[\omega(k)t - kr]), \tag{23}$$

where the frequency $\omega(k)$ is related to the wave vector $k$ through the dispersion relation

$$\omega(k) = \frac{1}{2}\left(k^2 + \delta\right). \tag{24}$$

These are right moving solutions, and the left moving ones can be obtained from the expressions above by replacing $k$ by $-k$ in the expression of $z$ [Eq. (23)].

Eq. (22) being linear, a general solution can be constructed by superposition of these plane waves:

$$z(r,t) = \int_{-\infty}^{\infty} \frac{dk}{2\pi} z_0(k) \exp(i[\omega(k)t - kr]), \tag{25}$$

where $z_0(k)$ is the Fourier transform of the initial condition at $t = 0$. Consider now the long-time and long-distance limit, with $x = r/t$ fixed:

$$z(r,t) = \int_{-\infty}^{\infty} \frac{dk}{2\pi} z_0(k) \exp(it[\omega(k) - kx]). \tag{26}$$

When $t \to \infty$ the integral above is dominated by the vicinity of a saddle point located at a wave vector $k_0$ determined by

$$\omega'(k_0) \;=\; x, \tag{27}$$

which gives

$$k_0 = x = r/t. \tag{28}$$

The saddle point integration then gives

$$z(r,t) \simeq \sqrt{\frac{2i\pi}{t}} \frac{z_0(k_0)}{2\pi} \exp\left[-ir^2/(2t) + it\delta/2\right]. \tag{29}$$

It is interesting to note that the complex phase in the equation above is independent of the initial condition; it describes (small) oscillations around the $z$ direction with an angle (in the

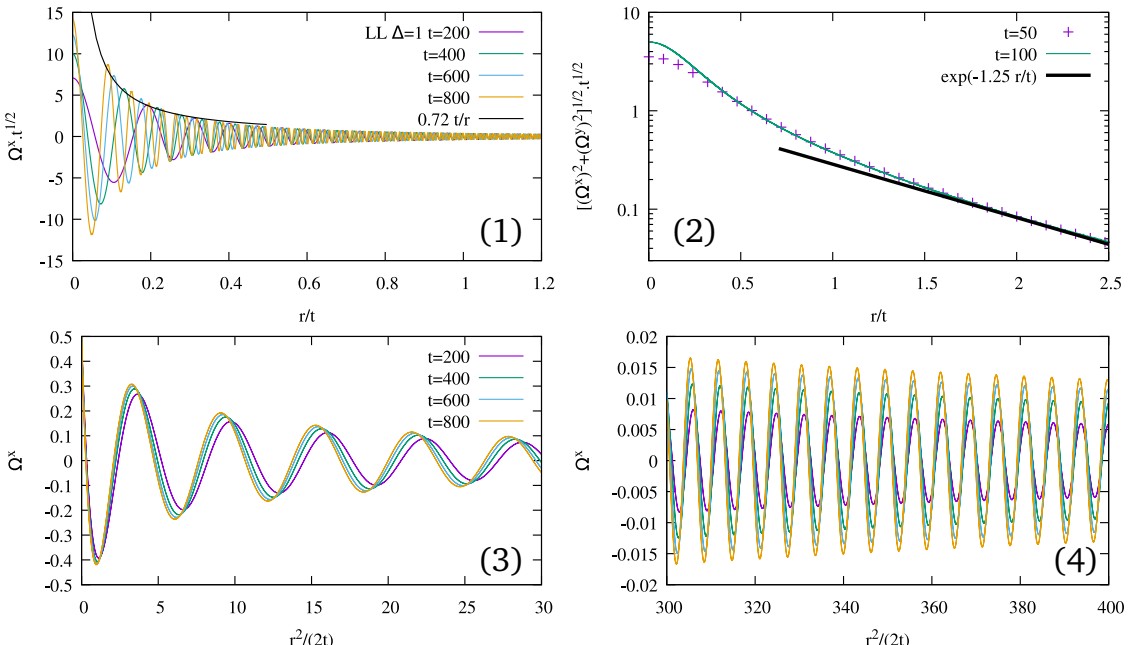

Figure 4: Panel 1 : $\Omega^x \sqrt{t}$ for $\Delta = 1$ as a function of $r/t$ (LL). With this scaling the different curves ($t = 200, 400, 600$ and $800$) appear to have the same envelope. The thin black line is a fit of the envelope in the region where $r$ is of the order of $\sqrt{t}$ (thus small $r/t$). In this region, the envelope of $\Omega^x \sqrt{t}$ appears to scale as $\sim t/r$. Panel 2: $\sqrt{(\Omega^x)^2 + (\Omega^y)^2} \sqrt{t}$ in log scale. The thick black line is a fit in the region of large $r/t$, where the amplitude of the magnetization in the $x - y$ plane decays exponentially. This plot suggests that $\sqrt{(\Omega^x)^2 + (\Omega^y)^2} = \sqrt{1/4 - (\Omega^z)^2}$ scales as $\exp(-cr/t)/\sqrt{t}$ for large $r/t$, which is consistent with Fig. 6. Panels 3 and 4: $\Omega^x$ as a function of $r^2/(2t)$. With this scaling, the period of the oscillations is almost the same for the four curves, and is found to be numerically close to $2\pi$. This behavior is observed both for small $r^2/t$ (panel 3), as well as large $r^2/t$ (panel 4). $\Omega^x$ can thus be described by an oscillatory factor $\cos(r^2/(2t) + \mathrm{cst})$, in agreement with Eq. (29).

$x$-$y$ lane) varying as $\varphi(r, t) = -r^2/(2t) + t\delta/2$. This is in agreement with the data plotted in Fig. 4 (bottom panel) for $\Delta = 1$ ($\delta = 0$) or Fig. 5 at $\Delta = 1.2$.

The initial condition given by Eq. (8) does not correspond to a small $|z|$ close to the origin at $r = 0$, but let us nevertheless consider the dynamics obtained with the linear equation [Eq. (22)], starting from this initial condition. The Fourier transform $z_0(k)$ of the initial condition can be computed explicitly, and also takes the form of the inverse of an hyperbolic cosine:

$$z_0(k) = \frac{1}{2} \int_{-\infty}^{\infty} \frac{dr}{\cosh(ar)} \exp(ikr) = \frac{\pi}{2a} \frac{1}{\cosh(\pi k/(2a))}, \tag{30}$$

plugging the equation above into Eq. (29) gives

$$z(r, t) \quad \simeq \quad \sqrt{\frac{2i\pi}{t}} \frac{1}{4a \cosh(\pi r/(2at))} \exp\left[-ir^2/(2t) + it\delta/2\right]. \tag{31}$$

In this linear approximation, the amplitude $|z|$ appears to be a function of $r/t$. This is in agreement with the upper panels of Fig. 4, which show that the envelope of the oscillations in $\Omega^x \times \sqrt{t}$ is a function or $r/t$. The analytical result shows that this amplitude scales as $|z| \sim \exp(-\pi r/[2at])/\sqrt{t}$ for large $r/t$, which is also in agreement with the data (upper right panel of Fig. 4). The numerical value of the constant $c \simeq 1.25$ obtained by a fit is however

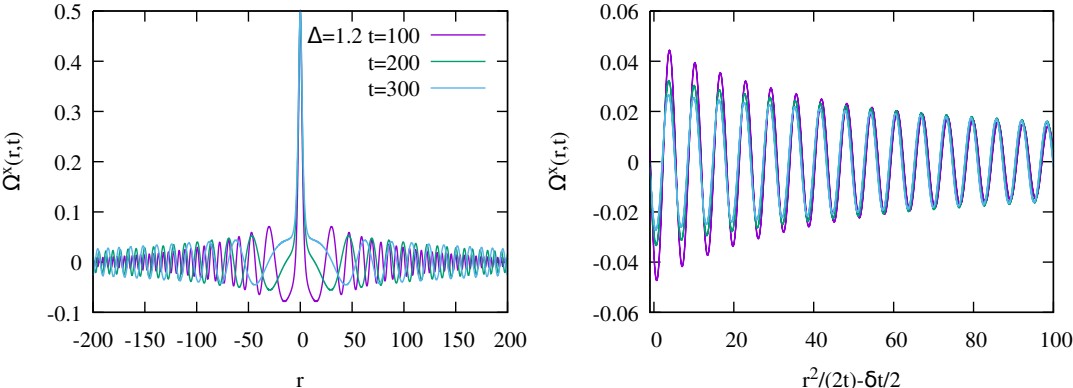

Figure 5: Large distance behavior of $\Omega^x(r, t)$ for $\Delta > 1$ and three different times. Left panel: $\Omega^x(r, t)$ as a function of the position $r$ for $\Delta = 1.2$. Right: Same data plotted as a function of $r^2/(2t) - t\delta/2$ with $\delta$ given in Eq (6). The latter shows that the phase of the oscillations is well described by the argument of the exponential in Eq. (29).

different from the constant $c' = \pi/(2a) = \pi/4$ predicted by the analytical argument above. This quantitative discrepancy for the rate of the exponential decay is presumably due to the fact that the initial condition is not in the linear regime close to the origin.

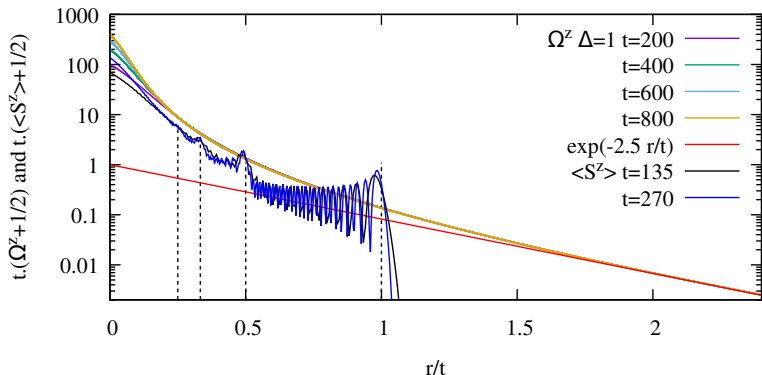

Figure 6: Zoom on the tail of the profile for $\Delta = 1$. For the LL model, we plot the quantity $t \cdot (\Omega^z + 1/2)$. The almost perfect collapse of the curves for different times shows that, at long times and large distances, we have $\Omega^z + 1/2 \simeq \frac{1}{t} \exp(-cr/t)$. For XXZ the black and blue lines correspond to the quantity $\langle S_r^z \rangle + \frac{1}{2}$ in the Heisenberg chain (tDMRG) at $t = 135$ and $t = 270$ (the two curves are almost on top for of each other). The data are consistent with the scaling observed for LL. The vertical dotted lines are at $r/t = \frac{1}{4}, \frac{1}{3}, \frac{1}{2}, 1$, and correspond to the maximum group velocities of bound-states of 4, 3, 2 and 1 magnons.

The modulus $|z|$ can also be extracted from the $z$ component of the magnetization, since $M^z = -\sqrt{1 - |z|^2}$. Fig. 6 indeed shows an exponential decay which is compatible with the result of the perturbative expansion. But importantly, the quantum chain shows a similar behavior. Although the latter displays additional oscillations [44], the tail of the $z$-magnetization profile exhibits some exponential-like decay in $r/t$ (black and blue lines in Fig. 6 correspond to different times and are almost on top of each other). This decay is however cut off at $r/t \simeq 1$, because of the maximal velocity in the chain. A qualitatively similar behavior is also observed in the XXZ chain for $\Delta > 1$ (data not shown). This exponential decay is however cut for

$r/t > 1$, which corresponds to the largest velocity in this lattice model: the maximum group velocity of an "up" spin in a background of down spins is indeed equal to unity. Similarly, we observe some "bumps" in the tail of the magnetization profile at $r/t \simeq \frac{1}{4}, \frac{1}{3}, \frac{1}{2}$, and these correspond to the maximal group velocities of bound-states of 4, 3 and 2 magnons propagating in the vacuum (*i.e.*, down spins) for $\Delta = 1$ [45]. Such bumps have also been observed in the Von Neumann entropy profiles [26]. Of course these are lattice effects and have no analog in the continuum.

# 6 Easy axis regime $\Delta > 1$

In the early study of the XXZ domain wall problem by Gobert *et al.* [24], the current $I(t) \equiv I(r = 0, t)$ associated to the transfer of the $z$-component of the magnetization (from the left half to the right half of the chain) was observed to vanish at long time for $\Delta > 1$. This absence of spin transport is consistent with exact Bethe ansatz results for the return probability (or Loschmidt echo) [25]. As for the LL problem, the $z$ magnetization profile was observed to freeze at long times [27, 28], and to be described by a soliton solution of the easy-axis LL equation. This was explained analytically by the fact that the associated inverse scattering problem involves a stable static kink in the easy-axis case [27].

In this section we show that, in the limit $\Delta \to 1^+$ both problems have the *same* soliton-like stationary magnetization profiles, with a spatial width scaling as $\delta^{-1/2}$ and which thus diverges when $\Delta \to 1^+$. The oscillations about the stationary profile appear to have a period proportional to $\delta^{-1}$, which can be understood from a dimensional analysis of easy-axis LL equation. Finally, as discussed in Sec. 5, at larger distances the perturbative analysis still holds and small amplitude oscillations around the $z$ axis propagate much further than the soliton width, but decay exponentially in $r/t$.

## 6.1 Numerics for LL and XXZ

The $z$ component of the magnetization profiles is plotted in Fig. 7 for a few values of $\Delta > 1$. The width of the stationary (or mean) profile grows when approaching the isotropic point. It nevertheless remains relatively limited even at $\Delta = 1.05$, where it does not exceed 20. In addition to the absence of propagation at long times, we observe oscillations with an amplitude which grows when $\Delta$ approaches 1. In the panel of Fig. 7 corresponding to $\Delta = 1.05$ we indeed see that the profile keeps changing up to $t = 400$. Remarkably we have a good quantitative agreement between the quantum chain and LL, without any adjustable parameter. Again, this should be viewed as a consequence of the energy conservation argument exposed in Sec. 3. The oscillations are easier to see in the current (defined in Eq. (5)), plotted in Fig. 8 for the LL and XXZ models. In this figure, the left panel shows some XXZ data for several values of $\Delta$ in the easy-axis regime. The current has some transient regime, after which it oscillates around zero. When multiplied by $\sqrt{t}$ (right panel of Fig. 8) the amplitude of the oscillations appear to be approximately constant. The data are thus compatible with a slow $t^{-1/2}$ decay of the oscillations, at least in the LL case. The situation for XXZ is slightly less clear and longer runs would be needed to establish firmly the form of the decay.

Concerning the temporal period of the oscillations, they appear to be proportional to $\delta^{-1}$. This can be seen in the middle panel of Fig. 8, where an approximate collapse of the current oscillations is achieved using a rescaled time $\tau = \delta t$. For the XXZ data at different values of $\Delta$, as well as for an easy-axis LL solution. Here again, we get a reasonable agreement between the classical and quantum models, without any adjustment.

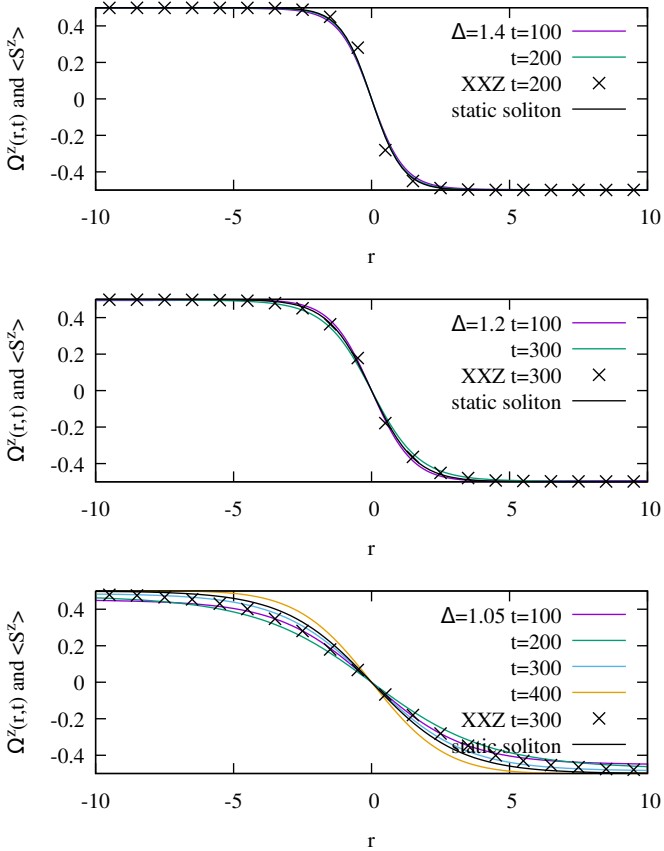

Figure 7: $\Omega^z(r, t)$ and $\langle S^z(r, t) \rangle$ as a function of the position $r$ for $\Delta = 1.4$ (top), $\Delta = 1.2$ (middle) and $\Delta = 1.05$ (bottom) and for different times from $t = 100$ to $t = 400$. The LL solution shows some long-lived oscillations around some mean profile. These are visible (larger amplitude) for $\Delta = 1.05$. The full black line represents the static soliton of Eqs. (32). Black crosses: expectation value $\langle \hat{S}^z \rangle$ in the quantum XXZ chain (tDMRG simulations). To visualize the amplitude and the period of these oscillations, see Fig. 8.

## 6.2 Easy-axis LL soliton

The observation made in the previous paragraph can be explained from the LL point of view, using the rescaled equation Eq. (9) and the rescaled time $2\tau = |\delta|t$.

One can easily check (see, e.g., Ref. [36]) that the following profile is an exact static solution when $\Delta > 1$ ($\delta > 0$):

$$2\Omega^x(r) = 1/\cosh\left[r\sqrt{\delta}\right] \tag{32a}$$

$$\Omega^y(r) = 0 \tag{32b}$$

$$2\Omega^z(r) = -\tanh\left[r\sqrt{\delta}\right]. \tag{32c}$$

It is a topological soliton, which lies in the $x$-$z$ plane. It corresponds to Eqs. (8) with $a = \sqrt{\delta}$, but this is not the value $a(\Delta)$ which ensures that the energy is that of the domain wall of the XXZ chain (see Sec. 3.2). The energy density of this solution is $\epsilon(r) = \frac{1}{4}\delta\left[\cosh(\sqrt{\delta}x)\right]^{-2}$ and its total energy is $E_S = \frac{1}{2}\sqrt{\delta}$. This energy is always lower than the XXZ energy $E_{\text{XXZ}} = \frac{1}{2}\Delta$.

Nevertheless, the numerical results of Figs. 7-8 show that the LL and XXZ profiles oscillate around, and then converge to, the soliton solution above. This convergence becomes very slow

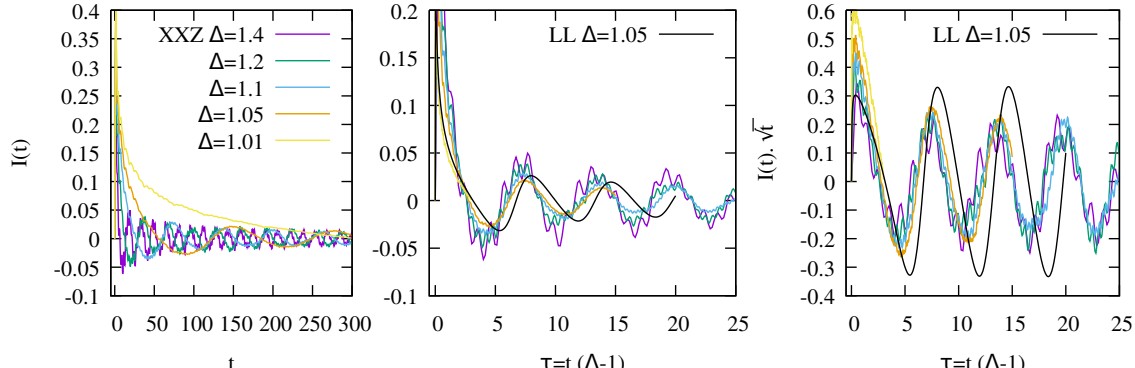

Figure 8: Left panel: Current $I(t)$ for different values of $\Delta$, from 1.01 to 1.4, in the XXZ chain. Data obtained from tDMRG simulations on a chain with $L = 600$ or $800$ sites and maximum bond dimension between 1000 and 2000. Middle panel: same data plotted as a function of $\tau = (\Delta - 1)t$. The relatively good collapse of the curves associated to different values of the anisotropy can be understood in the framework of the easy-axis LL model, thanks to the rescaling defined in Eqs. (9) which absorbs the anisotropy $\delta = 2(\Delta - 1)$ into a definition of a new time variable $\tau = \frac{1}{2}|\delta|t$. For comparison, the current obtained from the LL equation at $\Delta = 1.05$ is also shown (black line). Right panel: same data multiplied by $\sqrt{t}$, which gives oscillations with an approximately constant amplitude.

as $\Delta$ approaches 1. So, as argued in Ref. [27], we find that the excess energy is radiated away and that the central region of the system approaches the static soliton configuration. The scaling in Eq. (32) then naturally explains the observed width of the steady profile, proportional to $\delta^{-1/2}$. The rescaled time in Eq. (9) shows that the natural time scale is $\sim \delta^{-1}$, as the observed period of the oscillations in $I(t)$, both for LL and XXZ.

# 7 Isotropic model $\Delta = 1$

For the quantum chain, the domain wall problem in the isotropic case has already attracted a lot of attention. Although some studies [24,46] concluded that the expansion of $\langle S^z \rangle$ could be superdiffusive, a diffusive behavior with a logarithmic correction now seems more likely [26, 30] (still, some anomalous diffusion is possible when starting from a different initial condition that is close to an equilibrium state [46–49]). The same behavior was recently found of the isotropic LL model [27], and we provide below a detailed quantitative comparison of quantum spin chain with the LL model. This section being relatively long, we begin by a summary of the results.

First, we confirm numerically the findings of Ref. [27] concerning log-corrected diffusion of the $z$ component of the magnetization in LL. But we go further and provide a detailed analysis of the $x$ and $y$ components, the energy density, and the energy current in particular. We find that, both for the quantum spin chain and the LL problem, two regimes should be distinguished in the long time limit. First, a region where $r/\sqrt{t}$ (up to $\ln t$ factors) is finite, where the $z$ component is different from $\pm\frac{1}{2}$, and where the energy density is independent of $r$ and proportional to $\ln(t)/t$. In this region the magnetization can be described by a self-similar solution with an effective time-dependent energy density. We also characterize the regime of $r/t$ finite, which may be called the ballistic regime. There, $\langle S^z \rangle$ as well as $\Omega^z$ approach $\pm\frac{1}{2}$ with corrections which scale as $\sim \exp(-cr/t)$. $r/t$ appears to be the appropriate scaling variable

for these quantities, as well as for the energy or the energy current, contrary to what the behavior of the $z$ component could naively suggest. We also discuss an important difference between the quantum and the LL problems: the quantum state is invariant under rotation about the $z$ axis (hence $\langle S_r^x \rangle = \langle S_r^y \rangle = 0$), while the classical state is not. This symmetry difference enables us to conjecture some logarithmic growth of the entanglement entropy in the quantum chain (Sec. 7.2). We could also find a counterpart of the $x$ and $y$ components of the LL magnetization in the quantum side by using two-point correlations (Sec. 7.3). We also discuss (Sec. 7.5) the point of view where the LL magnetization $\vec{M}$ is the tangent vector of a curve in the three dimensional space. As an interesting finding, we observe that the torsion of this curve is remarkably simple and equal to $r/t$ at long times, in the diffusive as well as in the ballistic regime. We finally explain some of these findings by exploiting the mapping from the isotropic LL equation to the NLS equation, and constructing a variational solution to the latter.

## 7.1  $z$ component: current and diffusion with logarithmic correction

Solving numerically the isotropic ($\Delta = 1$) Landau-Lifshitz equation

$$\vec{\Omega}_t = \vec{\Omega} \wedge \vec{\Omega}_{rr}, \tag{33}$$

with the initial condition (8) reveals some diffusive behavior for the expansion of the $\Omega^z$ profile, with some (multiplicative) logarithmic correction. This confirms the finding of Ref. [27].

This can be seen by analyzing the current $I(t)$, as summarized in Figs. 9 and 10. For a conventional diffusive behavior, $I(t)$ would decrease as $t^{-\frac{1}{2}}$, and $I(t)\sqrt{t}$ would converge to a constant at long times. We instead see (left panel of Fig. 9) that it does not saturate, at least up to $t = 800$. The data are then fitted to (a) a constant minus a $1/\sqrt{t}$ correction, (b) a power low (*i.e.* superdiffusion) and (c) to a logarithm. The latter offers the best fit, and agrees with the LL solution far beyond the time interval $[300, 400]$ where the fit was performed (interval marked by the vertical dotted lines).[4] As can be seen in the right panel of Fig. 9, the agreement can be further improved by adding an additional $1/t$ correction [curve (d)].

Since the form $I(t) \sim \ln(t)/\sqrt{t}$ offers the best match with the data, this Ansatz for the current can be integrated over time to give the transfered magnetization from the left to the right since the initial time, and the result is $g(t) \simeq \sqrt{t}(\ln(t) + a)$. As shown in the bottom panel of Fig. 10, once the magnetic profiles $\Omega^z(r, t)$ and $\langle S_r^z(t) \rangle$ are plotted as a function of $r/g(t)$, the data obtained at different times collapse onto the same curve. The collapse is in particular much better than if one simply plots $\langle S_r^z(t) \rangle$ (or $\Omega^z(r, t)$) as a function of $r/\sqrt{t}$, as done in the upper panel of Fig. 10.

The stationary profiles obtained in the bottom of Fig. 10 appear to be slightly different for the LL problem and the quantum spin chain. From the conjectured asymptotic equivalence of the two problems one could have expected to see the very same profile in the two cases. But instead we may attribute the discrepancy to some memory of the initial state and of the short time evolution, where the energy density is finite and the two problems still inequivalent.

## 7.2  $U(1)$ symmetry and entanglement entropy in the XXZ chain

The states described by the LL equation break the rotational invariance about the $z$ axis. This is particularly obvious for the initial condition (8) where $\Omega^x(r = 0, t = 0) = \frac{1}{2}$ and $\Omega^y(r, t = 0) = 0$. On the other hand, the domain wall state used as an initial condition for the quantum chain is invariant under rotations about the $z$ axis, and the state $|\psi(t)\rangle$ therefore remains an eigenstate of $\hat{S}_{\text{tot}}^z = \sum_r \hat{S}_r^z$ at any time.

---

[4]Note that (c) is equivalent to Eq. (7) in Ref. [26], and was advocated to describe the behavior of the domain wall problem in the quantum Heisenberg chain.

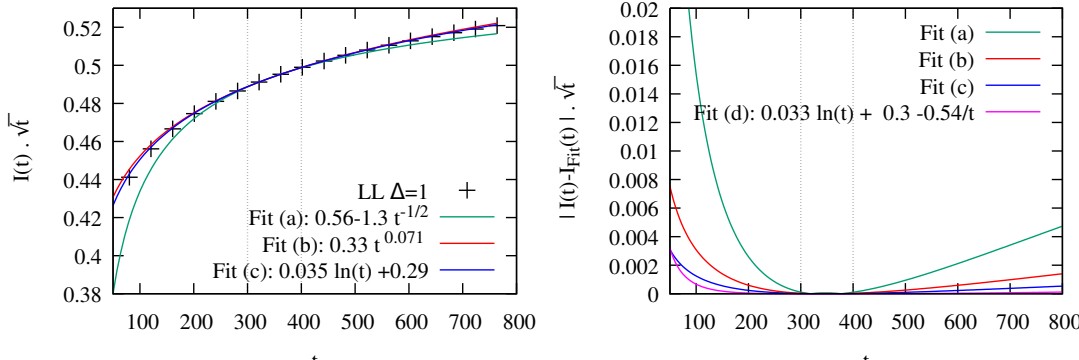

Figure 9: Left: LL current $I(t)$ [Eq. (5)] multiplied by $\sqrt{t}$ as a function of time, for $\Delta = 1$. The numerical solution is compared with three fitting functions. These fits are done using only the interval $[300, 400]$ (dotted vertical lines), and the quality of these fits can be judged by the discrepancy between the data and the fitting functions outside this interval. Fit (a) corresponds to $I(t) \simeq \sqrt{t} + cst/t$. Fit (b) represents a superdiffusive behavior $I(t) \simeq t^\alpha$ with $\alpha > 0.5$. The third one, (c), corresponding to $I(t) \simeq \ln(t)/\sqrt{t}$, appears to provide the best fit, and is in agreement with the results of Ref. [27]. The right panel shows the absolute value of the difference between the LL solution and the three fitting functions. Among the three fits (a),(b) and (c), the fit (c), which corresponds to a diffusive behavior with a multiplicative logarithm shows the best agreement with the numerical solution of the LL equation. Fit (d): including an additional $1/t$ correction to the function of fit (c) makes the agreement almost perfect at the scale of this plot.

We may thus take the classical LL state, and the associated magnetization vectors for all the integer values of the position $r$, and promote it to a (quantum) product state $|\psi(t)_{\text{LL}}\rangle$ for the spin chain. In order to restore the $U(1)$ symmetry, we can rotate it (globally) about the $z$ axis, and perform a linear combination of different rotated copies of this state. The resulting state could then be written a $|\psi(t)_{\text{LL,sym}}\rangle = \int_0^{2\pi} d\theta \exp\left(i\theta \hat{S}^z_{\text{tot}}\right)|\psi(t)_{\text{LL}}\rangle$ [50, 51]. It is then clear that this $U(1)$-symmetric state is no longer a product state. The symmetrization thus introduced some quantum entanglement in the system.

How can we estimate the Von Neumann entropy associated to a left-right partition on the chain ? We should first estimate the number of linearly independent states which are generated by the rotations above. For a macroscopic quantum spin of large size $S$, the rotations would span a space of dimension $\mathcal{O}(S)$ [50, 51]. We may use a similar argument for $|\psi(t)_{\text{LL}}\rangle$, which we assume here to be nontrivial only over a spatial region $[-R(t), R(t)]$, and which is therefore a state involving different spin quantum numbers up to $S_{\text{max}} \sim \mathcal{O}(R(t))$. We may thus consider that the symmetrized state $|\psi(t)_{\text{LL,sym}}\rangle$ is actually a linear combination of $\mathcal{O}(R(t))$ states which are orthogonal to each other. In turn, we may take the log of this dimension to estimate the von Neumann entropy. For large $R(t)$ we would thus have $S_{\text{vN}} \sim \ln(R(t))$.

We plot in Fig. 11 the entanglement entropy up to $t = 400$ at $\Delta = 1$, compared with $\frac{1}{2}\ln(t)$. The latter would be the leading term in the entropy generated by symmetrization for a region of length $\sim \sqrt{t}$, corresponding to diffusion.[5] As already mentioned in Ref. [26], the entangle-

---

[5] The present argument does not apply away from $\Delta = 1$. In particular for $\Delta < 1$ the nearest-neighbor correlations deviate from those of a perfectly ferromagnetic state (by an amount which is $\mathcal{O}(|\delta|)$, see Sec. 3), even at long times. Although these small deviations may only weakly affect the local magnetization or the energy density, they change completely the entanglement entropy. This can be illustrated, for instance, by considering a much simpler case: the homogeneous ground-state of the xx chain, with a magnetization per site $m$ close (but not equal) to $-1/2$.

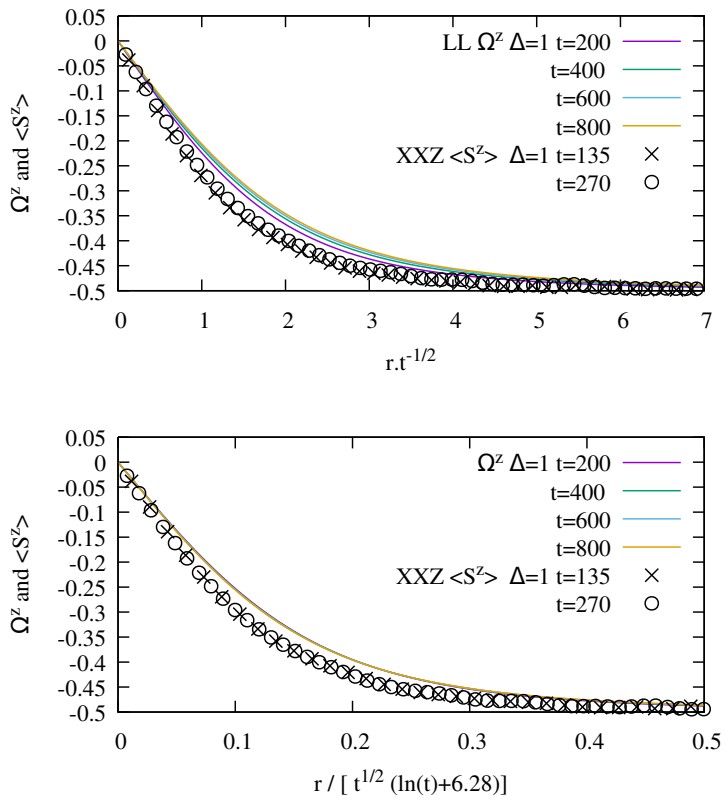

Figure 10: $\Omega^z$ magnetization profile for LL at $\Delta = 1$, and comparison with $\langle S^z \rangle$ in the XXZ chain (tDMRG simulations $L = 800$ sites, maximal bond dimension $\chi = 2000$ and time $t = 270$). The horizontal axis is $r/\sqrt{t}$ in the top panel, and $r/g(t)$ with $g(t) = \sqrt{t}\,[\ln(t) + 6.28]$ in the lower panel. The latter, gives a good collapse of the curves obtained at different time. The same multiplicative logarithmic correction to a diffusive behavior was also observed in Ref. [26] for the XXZ chain and in Ref. [27] for LL. Remark: the constant 6.28 in $g(t)$ is equivalent to the function (c) in Fig. 9, since $0.07 \frac{d}{dt}\left[(\ln(t) + 6.28)\sqrt{t}\right] = [0.29 + 0.035\ln(t)]/\sqrt{t}$.

ment entropy data appear to be compatible with $\sim \ln(t)$, but, with the times accessible to our simulations, the coefficient of the logarithm seems slightly larger than 0.5 (see fits (A) and (A') in Fig. 11). But we should stress that a power-law behavior can also not be excluded [26,46]. In particular, the present data appear to be well described by $S_{vN} \sim t^{0.16}$, as shown by the fit (B') in the bottom panel of Fig. 11. In any case, $\frac{1}{2}\ln(t)$ is only the semiclassical contribution to the entanglement, and it appears that some other contribution(s) to the entanglement entropy need to be included, like for instance that carried by the ballistic magnons (and bound states of magnons) in the region where $r/t$ is finite. We leave for future studies the task of determining the precise law of entanglement growth in this problem at $\Delta = 1$, but the $U(1)$ symmetry argument above should provide a starting point.

---

The spin-spin correlations can be made arbitrarily close to that of a ferromagnet by taking $m$ to $-1/2$. Nevertheless the quantum state is a (low but finite density) Fermi sea. In such a case the entropy of a long segment of length $l$ scales as $\sim \log(l)/3$. This is clearly very different from the situation with $m = -1/2$ exactly, which is a product state. Going back to the domain wall problem, the free fermion case $\Delta = 0$ indeed gives $S_{vN} \sim \frac{1}{6}\ln(t)$ [52,53]. As for tDMRG results (data not shown) at $\Delta = 0.5$ and $0.7$ (up to $t \sim 200$) suggest that this scaling may hold for all $\Delta < 1$.

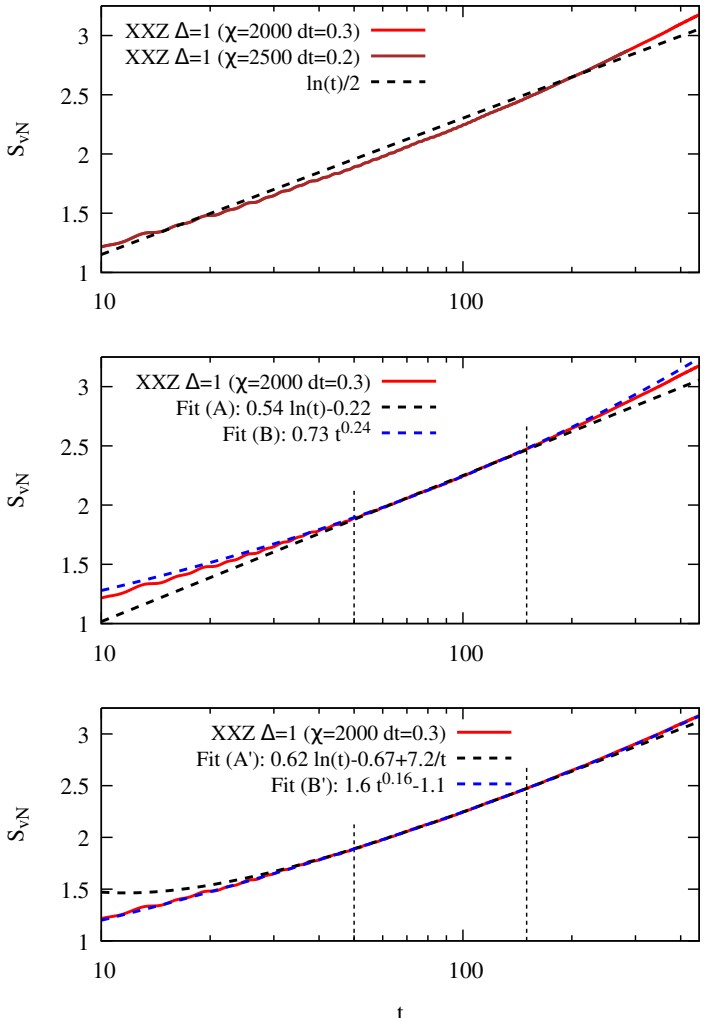

Figure 11: Von Neumann entanglement entropy for the quantum model at $\Delta = 1$, for a partition in the center of the chain. Top: tDMRG calculations with $L = 800$, time step $dt = 0.2$, maximum bond dimension $\chi = 2500$, MPS truncation parameter $10^{-11}$, with time up $t = 286$ (brown line). At the scale of the plot the results are unchanged when taking $dt = 0.3$, $\chi = 2000$ and the truncation parameter to $10^{-10}$ (data shown up to $t = 400$, red line), which indicates a good convergence. We also show $\frac{1}{2}\ln(t)$ (see Sec. 7.2), which however underestimates the entropy growth at large times. Middle: two-parameter fits of the entanglement entropy, with (A) $a\ln(t)+b$ or (B) a power law $c\,t^d$. Bottom: three-parameter fits of the entanglement entropy, with (A') $a\ln(t)+b+c/t$ or (B') $c\,t^d+e$. The fit (B'), with an exponent close to $1/6$, seems to offer the best description of the data, but a logarithmic growth cannot be excluded. All the fits are performed in the interval $[50, 150]$ (dashed vertical lines).

## 7.3 Correlations in the quantum chain and $x$ and $y$ components of the LL magnetization

As explained above, there is a $U(1)$ symmetry in the spin-$\frac{1}{2}$ chain, and this implies that $\langle \hat{S}_r^x \rangle = \langle \hat{S}_r^y \rangle = 0$ at any time. On the other hand, the states described by the LL equation break the rotational invariance about the $z$ axis.

It is nevertheless possible to compare $\Omega^x$ or $\Omega^y$ computed in the LL framework with data from the quantum chain. To do so, one has to consider the two-point spin-spin correlations in the quantum chain, like $\langle \hat{S}_0^x \hat{S}_r^x \rangle$. One can obviously write $\hat{S}_0^x = \hat{P}_x - \frac{1}{2}$, where $\hat{P}_x = \hat{S}_0^x + \frac{1}{2}$ projects onto states where $S_0^x = \frac{1}{2}$. Thanks to the rotational invariance mentioned above, we have $\langle \hat{S}_0^x \hat{S}_r^x \rangle = \langle \hat{P}_0^x \hat{S}_r^x \rangle$. Since $P_0^x = 1$ with probability $\frac{1}{2}$ (again due to rotational invariance), and zero otherwise, it is natural to compare $\langle \hat{S}_0^x \hat{S}_r^x \rangle$ with $\Omega^x$ in the LL problem, where, by construction, $\Omega^x(r = 0, t) = 1/2$. Concerning the normalization, consider the ferromagnetic state of $N$ spins $1/2$ that has $S_{\text{tot}}^z = 0$ (but total spin $S_{\text{tot}} = N/2$). Such a state is the symmetrized version of the ferromagnetic state pointing in the (say) $x$ direction. In such state one can easily compute the spin-spin correlations: $\langle \hat{S}_0^x \hat{S}_r^x \rangle = \langle \hat{S}_0^y \hat{S}_r^y \rangle = \frac{N}{8(N-1)}$ for $r \neq 0$, which approaches $\frac{1}{8}$ for a large system. We should then compare $4\langle \hat{S}_0^x \hat{S}_r^x \rangle$ with $\Omega^x(r, t)$, and similarly, $4\langle \hat{S}_0^x \hat{S}_r^y \rangle$ with $\Omega^y(r, t)$.

The result is displayed in Fig. 12. First we see in the upper panel that $4\langle \hat{S}_0^x \hat{S}_r^x \rangle$ approaches $\frac{1}{2}$ when $r \to 0$, in agreement with the argument above and the fact that, in the center of the system and at long times, the quantum chain resembles locally a symmetrized ferromagnetic state with $\langle S_r^z \rangle = 0$. For larger $r$ we observe some some semi-quantitative agreement between the quantum and classical models, without any adjustable parameter. In particular, the period of the spatial oscillations in the correlations of the quantum systems is quite close to the period of the oscillations in the LL magnetization. An even better agreement between correlations in the spin-$\frac{1}{2}$ chain and in-plane magnetization of the LL problem is shown in the bottom panel of Fig. 2 in an easy-plane case. The amplitude of the correlations in Fig. 12 appears however to decay faster than the LL magnetization. We note that this may be due in part to the fact, for $r > 0$, the $\langle \hat{S}^z \rangle$ profile lies slightly below $\Omega^z$, as can be seen in Fig. 10. Another source of discrepancy is the fact that, in the quantum chain, the center of symmetry is in the middle of a bond, at $r = 1/2$, while it is at $r = 0$ in the LL model. Finally, it is plausible that some quantum effects remain here for this observable, even at long times.

For $r/t$ finite and $\Delta = 1$, the $z$ component of the magnetization tends to $\pm\frac{1}{2}$, both in the quantum and classical problems. This is a regime where the expansion of Sec. 5 applies and the oscillations of the $x$ and $y$ components of the magnetization in LL are described by an azimuthal angle $\varphi(r, t) = -\frac{r^2}{2t}$. This regime is realized in the right part of Fig. 12. On the other and, for $r$ of the order of $\sqrt{t}$ (or even $\sqrt{t} \ln t$), which corresponds to the first few oscillations in the left of Fig. 12, a different behavior of $\varphi$ is observed, as discussed in the next paragraph.

## 7.4 Energy density

For the isotropic LL model the energy density [Eq. (7)] is $\epsilon(r, t) = \frac{1}{2}\left(\vec{\Omega}_r\right)^2$. This quantity is displayed in Figs. 13 and 15. The top panel of Fig. 13 shows that, at least for sufficiently large $r/t$, the energy density multiplied by time is well described by a function of $r/t$: $\epsilon(r, t) = \frac{1}{t}K(r/t)$. For small $r/t$, the collapse of the energy curves associated to different values of $t$ is not good, and this indicates that $K$ is singular at the origin. The data plotted in Fig. 16 indicate a logarithmic divergence of $t \cdot \epsilon(r = 0, t)$ with time. From this we may infer that (in the limit of infinite time) the function $K$ exhibits a logarithmic divergence: $K(u) \sim |\ln u|$ at small $u = r/t$.

The energy density of the spin-$\frac{1}{2}$ Heisenberg model at $r = 0$, measured through $\left\langle \frac{1}{4} - \vec{S}_0 \cdot \vec{S}_1 \right\rangle$ on the central bond of the chain, is also represented in Fig. 16, as a function of time. It displays oscillations, but otherwise follows relatively well the LL behavior, namely $\epsilon(r = 0) \sim 0.06\left[\ln(t) + 2.4\right]/t$ (see the fitting function in Fig. 16).

In the right panel of Fig. 13 the spatial variations of the energy density for LL is compared to those of the Heisenberg chain. Although the energy density of the quantum chain shows some important oscillations (their scaling in the vicinity or $r/t = 1$ is discussed in [44]), it follows the LL data relatively well, and it certainly extends beyond the diffusive scale. It should

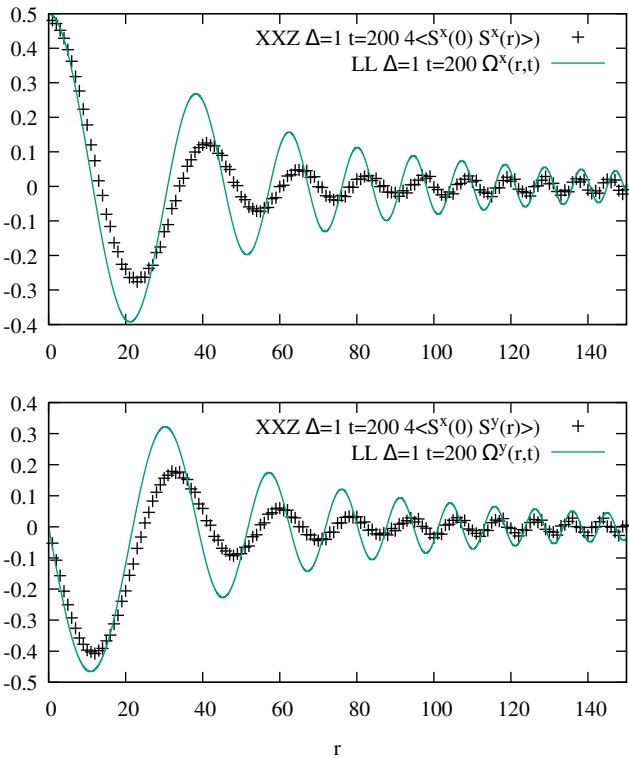

Figure 12: Comparison between some two-point correlations in the spin-$\frac{1}{2}$ chain with $\Omega^{x,y}(r,t)$ in the isotropic LL problem. Upper panel: $\Omega^x$ versus $\langle \hat{S}^x(0)\hat{S}^x(r \neq 0)\rangle$. Bottom panel: $\Omega^y$ versus $\langle \hat{S}^x(0)\hat{S}^y(r \neq 0)\rangle$. Both taken at time $t = 200$. See Sec. 7.3.

be noted that there is again no adjustable parameter here, except for the fact that the width the of LL profile at $t = 0$ was chosen to ensure a total energy equal to that of the quantum chain (Sec. 3.2).

After averaging the short-distance oscillations, this energy density shows a maximum in $r/t \simeq 1$ and $r/2 \simeq 0.5$. As observed already in Fig. 6, these two velocities correspond respectively to the maximum group velocity of a single magnon, and to the maximum group velocity of a bound state of two-magnons. The effects of these discrete velocities are specific to the quantum chain and have no direct analog in the LL problem.

For the isotropic LL model, the energy density can be written using the polar angles [Eq. (14)] and their space derivatives:

$$\epsilon = \frac{1}{2}\left(\vec{\Omega}_r\right)^2 = \frac{1}{8}\left(\vec{M}_r\right)^2 = \frac{1}{8}\left[(\theta_r)^2 + (\sin\theta\,\varphi_r)^2\right]. \tag{34}$$

From Fig. 10 (bottom panel) we know that the spatial scale for the variation of $\Omega^z$ is $\sim \sqrt{t}\ln t$. As a consequence, the gradient $\theta_r$ must scale as $\sim \left(\sqrt{t}\ln t\right)^{-1}$ in this region. So, the observed logarithmic divergence in $t \cdot \epsilon(r = 0, t)$ is not associated to the $(\theta_r)^2$ term, but must come from the second term in Eq. (34). We conclude that the gradient of the azimuthal angle should scale as $|\varphi_{r|r=0}| \sim \sqrt{\ln t/t}$. Numerically we indeed find that $|\varphi| \simeq 0.22\pi r\sqrt{(\ln t + 2.5)/t}$ fits the data relatively well for small $r/\sqrt{t}$, corresponding roughly to one full rotation about the $z$ axis ($-\pi < \varphi < \pi$).

In the limit where $\Omega^z$ is close to $\pm\frac{1}{2}$ the energy density given in Eq. (34) is dominated by the term proportional to $(\varphi_r)^2$. From the perturbative expansion of Sec. 5 we have $\varphi = -r^2/(2t) + t\delta/2$ and $\varphi_r = -r/t$. Since, by definition, $\sin(\theta) = 2\sqrt{(\Omega^x)^2 + (\Omega^y)^2}$, we can

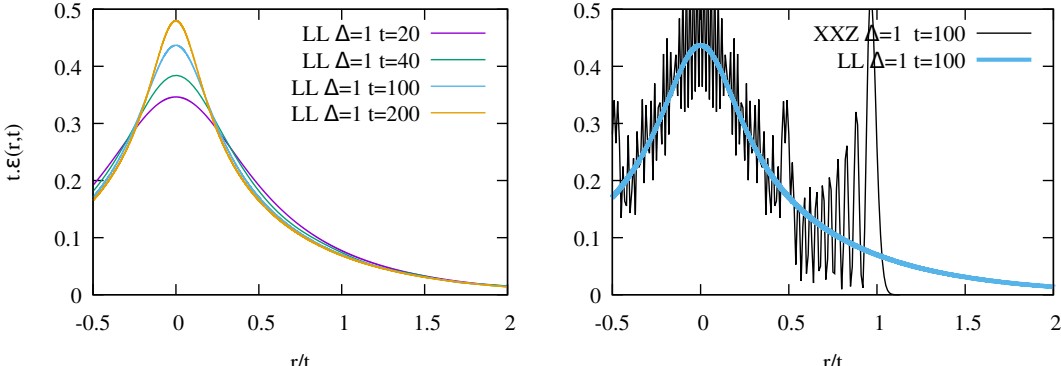

Figure 13: Left panel: LL energy density $\epsilon(r, t) = \frac{1}{2}\left(\vec{\Omega}_r\right)^2$ multiplied by $t$, as a function of $r/t$ for $\delta = 0$ (isotropic model). At sufficient large $r/t$, the curves associated to different times are practically on top of each other. The growth of $\epsilon(r = 0, t)$ with $t$ is compatible with a logarithmic time dependence ($\sim \ln t$ behavior, see Fig. 16). Right panel: The LL results are compared with XXZ data (black line) at $t = 100$ (longer times give a similar curves). For the latter there is a maximum group velocity equal to one, which explains why the "signal" is essentially zero for $|r/t| > 1$. Thanks to the energy conservation, the area of all these curves (LL and XXZ) is equal to $\frac{1}{2}$.

write the energy density in the perturbative approximation as:

$$\epsilon_{\mathrm{SW}} = \frac{1}{2}\left[(\Omega^x)^2 + (\Omega^y)^2\right](r/t)^2.\tag{35}$$

In other words, the ratio

$$\epsilon_{\mathrm{SW}}/\epsilon = \frac{\left[(\Omega^x)^2 + (\Omega^y)^2\right]r^2}{2t^2\epsilon}\tag{36}$$

should approach 1 when the perturbative approximation becomes asymptotically exact. This quantity is plotted in Fig. 14. As expected, the length scale beyond which the perturbative calculation becomes accurate is $\sim \sqrt{t \ln t}$, since this scale governs how $\Omega^z$ approaches $\pm\frac{1}{2}$.

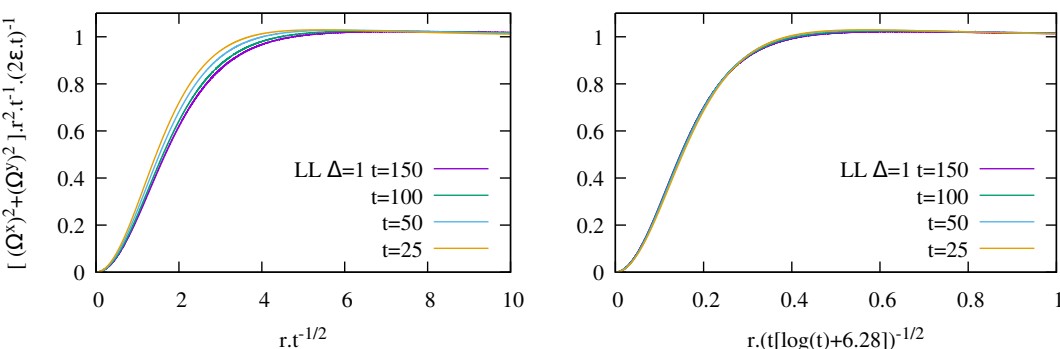

Figure 14: Ratio of the energy density computed in the perturbative approximation to the actual energy density. See text and Eq. (35) for details. The good curve collapse observed in the right panel indicates that this ratio obeys a scaling form similar to that of $\Omega^z$ (compare with the bottom panel of Fig. 10).

### 7.5 Vortex filament, torsion and self-similar solutions

As noted by Lakshmanan *et al.* [38], it is useful to interpret $r \to \vec{M}(r,t) = 2\vec{\Omega}(r,t)$ (at fixed $t$) as the tangent vector of a curve $\mathcal{C}$ parametrized by $r$. The LL energy density is then simply related to the curvature $\kappa$ of $\mathcal{C}$:

$$\kappa = ||\vec{M}_r|| = \sqrt{8\epsilon}. \tag{37}$$

Another important geometrical property is the torsion $\tau$ of $\mathcal{C}$ ($\tau$ should not to be confused with the rescaled time defined in Sec. 2.2). Using the LL variables it reads

$$\tau = \kappa^{-2}\vec{M} \cdot \left(\vec{M}_r \wedge \vec{M}_{rr}\right) = \epsilon^{-1}\vec{\Omega} \cdot \left(\vec{\Omega}_r \wedge \vec{\Omega}_{rr}\right). \tag{38}$$

This quantity is plotted in Fig. 18 for the LL solution we are interested in at $\Delta = 1$. The energy current $J = \vec{\Omega} \cdot \left(\vec{\Omega}_r \wedge \vec{\Omega}_{rr}\right)$ is also plotted in Fig. 17. The torsion data show that, at sufficiently long times, the LL solution is well descried by a remarkably simple relation $\tau \simeq r/t$. As discussed below, this is reminiscent of some particular solutions of the LL problem.

The equation of motion for the magnetization is Eq. (33), which is equivalent to $2\vec{M}_t = \vec{M} \wedge \vec{M}_{rr}$, or to the more conventional $\vec{M}_{\tilde{t}} = \vec{M} \wedge \vec{M}_{rr}$ with $2\tilde{t} = t$. The induced dynamics for the curve $\mathcal{C}$ is the so-called binormal flow [54], which describes the motion of a vortex filament in the local induction approximation. Some particular attention has been devoted to special solutions where $\vec{M}(r < 0, t = 0) = \vec{e}_z$ and $\vec{M}(r > 0, t = 0) = \cos(\gamma)\vec{e}_z + \sin(\gamma)\vec{e}_x$ [27, 38, 54–58]. This is an singular initial condition, where the curve $\mathcal{C}$ forms a sharp corner with angle $\gamma$ at $r = 0$. It then evolves to a smooth $\vec{M}(r,t)$ and $\mathcal{C}(t)$ for $t > 0$. This problem can be solved analytically for $\gamma < \pi$, and gives a self-similar solution where $\vec{M}(r,t)$ is a function of $r/\sqrt{t}$ (see Appendix C for an explicit expression of the self-similar solutions in terms of Kummer confluent hypergeometric functions). It is characterized by a spatially uniform curvature $\kappa = E/\sqrt{\tilde{t}} = E/\sqrt{t/2}$ and linear torsion $\tau = r/(2\tilde{t}) = r/t$ [38]. The constant $E$ is related to the corner angle through the relation $e^{-\pi E^2/2} = \cos(\gamma/2)$ [54]. Since $E$ diverges when $\gamma$ approaches $\pi$ – which is precisely the situation we are interested in – the self-similar solution breaks down in that limit. It is thus natural to trace back the appearance of a logarithmic correction to diffusion at $\gamma = \pi$ to this singularity in $E$ [27].

There is nevertheless a similarity between the self-similar solutions at $\gamma < \pi$ and our case (smooth initial condition and asymptotic angle $\gamma = \pi$): the energy density is $\sim \text{cst}/t$ for the self-similar solutions (at any $r/t$), while we showed that in our case that it is $\sim \ln t/t$ for $r \ll t$. But the torsion $\tau$ (Fig. 18) shows a more striking analogy, since we have $\tau = r/t$ in both cases, in the regime $r \sim \sqrt{t}$ as well as for $r/t \sim \mathcal{O}(1)$.

Fig. 19 shows a comparison between the smooth domain wall problem at $t = 200$ and the self-similar solution with the same energy density at $r = 0$ (or same curvature $\kappa$). This is achieved by taking the parameter $E$ of the self-similar solution to be $E^2 = 4t \cdot \epsilon(r = 0, t)$. The two states agree by construction when $r/\sqrt{t}$ is small, but differ otherwise. In particular, oscillations in the $z$ component are present in the self-similar solution, but absent from the LL we are interested in. We can also, by inspecting the behavior of the $M^y$ component, see that the asymptotic direction of the magnetization (when $r \to \pm\infty$) is not $\vec{M} = \pm\vec{e}_z$ in the self-similar profile. This due to the fact that $E$ is finite and therefore $\gamma < \pi$. Repeating this comparison for much larger $t$ would give a larger effective $E \sim \ln(t)$, an angle $\gamma$ closer to $\pi$, and a larger range of $r/\sqrt{t}$ where the the two solutions are close to each other. We thus find that one can describe the LL magnetization profile in the regime where $r/\sqrt{t}$ is of order one (but $r \ll t$) by self-similar solutions with an effective time-dependent parameter $E(t) \sim \ln(t)$. This can however not describe the system in the region where $r/t$ is finite, where the energy density is no longer constant.

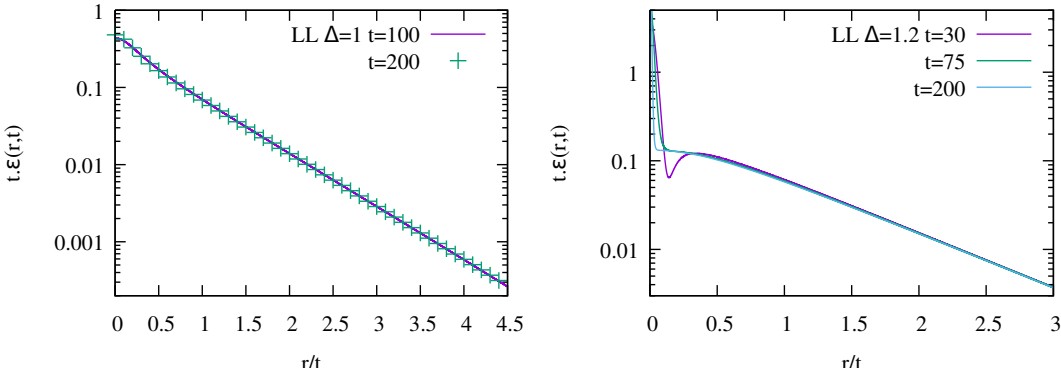

Figure 15: Left panel: LL energy density multiplied by $t$ in the isotropic case ($\Delta = 1$). Away from the center the energy density appears to decay as exponentially, $\epsilon(r,t)t \sim \exp(-cr/t)$ with $c \simeq 1.6$. See Sec. 7.7 for an explanation of this behavior (variational approach). Right panel: LL energy density multiplied by $t$ in an easy-axis case ($\Delta = 1.2$). At long times we again have $\epsilon(r,t)t \sim \exp(-c'r/t)$ for $r/t > 0$, but a different behavior for $|r| \ll t$ (see Sec. 6).

## 7.6 Nonlinear Schrödinger equation

The isotropic LL equation is well known to be equivalent to the Nonlinear Schrödinger (NLS) equation [59–61]. The mapping, also known as the Hasimoto transform, can be summarized as follows (more details in Appendix A). As explained above, we start from a solution of $2\vec{M}_t = \vec{M} \wedge \vec{M}_{rr}$, or equivalently $\vec{M}_{\tilde{t}} = \vec{M} \wedge \vec{M}_{rr}$ (with $2\tilde{t} = t$). One then constructs the associated curve $\mathcal{C}$ (the vortex filament) and obtains its curvature $\kappa(r,t)$ and torsion $\tau(r,t)$. Then one defines the following "filament function" (or NLS wave-function):

$$u(r,t) = \kappa(r,t)\exp\left[i\int_0^r \tau(r',t)dr'\right]. \tag{39}$$

On can then show that $u(r,t)$ satisfies a NLS equation:

$$iu_{\tilde{t}} = 2iu_t = -u_{rr} - \frac{1}{2}u\left(|u|^2 - \Lambda(t)\right), \tag{40}$$

where $\Lambda(t)$ only depends on time, and is explicitly given in terms of the curvature and the torsion close to the origin:

$$\Lambda(t) = \left(2\frac{(\tau^2)_{rr} - \kappa\tau^2}{\kappa} + \kappa^2\right)_{|r=0}. \tag{41}$$

Alternatively, one can define another wave-function $\psi$:

$$\psi(r,t) = u(r,t)\exp\left(\frac{i}{4}\int_0^t \Lambda(t')dt'\right), \tag{42}$$

which is then solution of a cubic NLS equation

$$2i\psi_t = -\psi_{rr} - \frac{1}{2}|\psi|^2\psi. \tag{43}$$

If the LL magnetization initially lies in a plane, the torsion vanishes and the corresponding NLS wave-function can be taken to be real at $t = 0$. We may thus write (at $t = 0$)

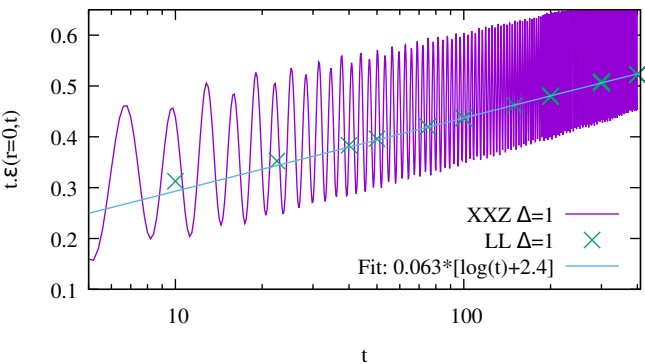

Figure 16: Energy density $\epsilon(r = 0, t)$ in the center of the system and multiplied by $t$, plotted as a function of $t$ (logarithmic scale). The green crosses correspond to the LL model and the purple curve corresponds to the quantum chain (both at $\Delta = 1$). The data are compatible with a logarithmic increase with time (see fitting function in the legend, blue line).

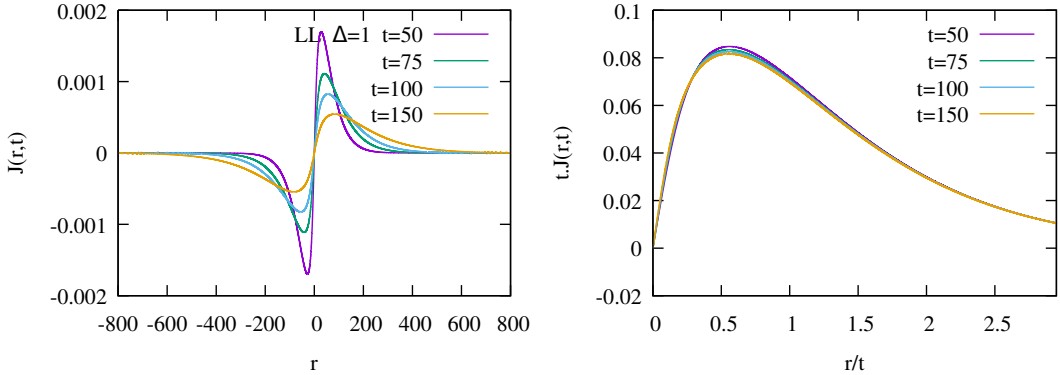

Figure 17: Left panel: LL energy current $J = \vec{\Omega} \cdot \left( \vec{\Omega}_r \times \vec{\Omega}_{rr} \right)$ at $\Delta = 1$, plotted as a function of the position $r$ for different times. Right panel: same data, multiplied by $t$ and plotted as a function $r/t$. An excellent collapse is observed for $r/t \gtrsim 1.5$.

$\vec{M} = (\cos(\theta(r)), \sin(\theta(r)), 0)$ and the curvature is given by $\kappa = |\theta_r|$. If we further assume that $\theta_r > 0$ everywhere, the filament function is simply given by $u = \theta_r$. As a consequence, the integral of $u$ gives the angle difference $\gamma$ between $r = -\infty$ and $r = \infty$:

$$\gamma = \int_{-\infty}^{\infty} u(r, t = 0) dr. \tag{44}$$

With the initial condition given in Eq. (8) and $a = 2$, we have $\kappa = 2/\cosh(2r)$ and

$$\psi(r, t = 0) = u(r, t = 0) = \kappa(r, t = 0) = \frac{2}{\cosh(2r)}. \tag{45}$$

This type of initial condition for the NLS equation was first analyzed by Satsuma and Yajima [62], using inverse scattering methods. From the above remark about the integral of $u$ one can check that $\gamma = \pi$. Importantly, for this precise value of the angle the inverse scattering approach is complicated by the presence of divergences [27] and we are not aware of an exact analytical treatment of this case. These divergences are also responsible for the logarithmically enhanced diffusion of the $z$ magnetization. One can however treat the case of an angle slightly below $\pi$ [27].

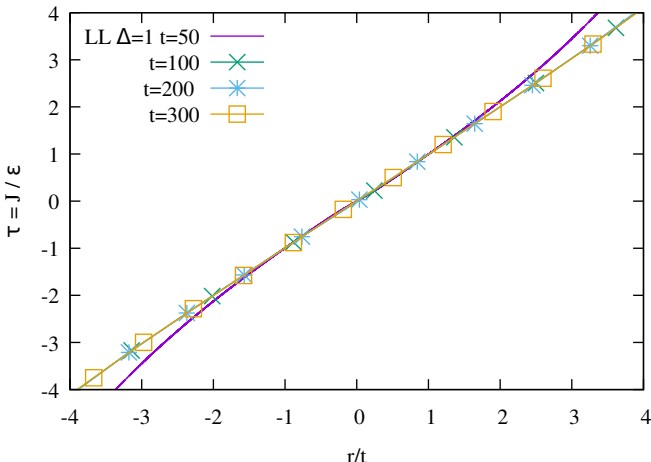

Figure 18: LL torsion $\tau$ plotted as a function of $r/t$. The good data collapse shows that, at long times, one simply has $\tau = r/t$.

## 7.7 Variational solution

In this section one considers the focusing NLS equation

$$i\psi_t = -\frac{1}{2m}\psi_{rr} - g\,|\psi|^2\psi\,, \quad \text{with} \quad m > 0 \quad \text{and} \quad g > 0\,. \tag{46}$$

Eq. (40) is recovered by taking $m = 1$ and $g = 1/4$. To facilitate comparisons with other conventions, we will however keep $m$ and $g$ general in the calculation below. Equation (46) is associated with the Lagrangian density

$$\mathscr{L} = \frac{i}{2}\left(\psi^*\psi_t - \psi\psi_t^*\right) - \frac{1}{2m}|\psi_r|^2 + \frac{g}{2}|\psi|^4 = -\text{Im}(\psi^*\psi_t) - \frac{1}{2m}|\psi_r|^2 + \frac{g}{2}|\psi|^4\,, \tag{47}$$

and has a soliton solution of the form

$$\psi_{\text{sol}}(r,t) = \frac{1}{\sqrt{gm}}\frac{A_0}{\cosh(A_0 r)}\exp\{iA_0^2 t/2m\}\,, \tag{48}$$

where $A_0$ is an arbitrary constant.

One considers a variational ansatz parametrized by 4 time-dependent quantities: $A(t)$, $B(t)$, $C(t)$ and $D(t)$:

$$\psi_{\text{var}}(r,t) = \frac{1}{\sqrt{gm}}\frac{B}{\cosh(Ar)}\exp\{i(C + Dr^2)\}\,. \tag{49}$$

Following the variational method proposed in Ref. [63], we insert the form (49) into the Lagrangian density (47). It yields a Lagrangian for the quantities $A$, $B$, $C$ and $D$:

$$\begin{aligned}
L &= \int_{\mathbb{R}} dr\,\left\{-(\dot{C} + \dot{D}r^2)|\psi_{\text{var}}|^2 - \frac{1}{2m}|\psi_{\text{var}}|^2\left(A^2\tanh^2(Ar) + 4D^2r^2\right) + \frac{g}{2}|\psi_{\text{var}}|^4\right\} \\
&= \frac{1}{3gm}\left(-6\frac{\dot{C}B^2}{A} - \frac{\pi^2}{2}\frac{\dot{D}B^2}{A^3} - \frac{1}{m}AB^2 - \frac{\pi^2}{m}\frac{B^2D^2}{A^3} + \frac{2}{m}\frac{B^4}{A}\right) \\
&= \frac{1}{3gm}\left(-6\dot{C}E - \frac{\pi^2}{2}\frac{\dot{D}E}{A^2} - \frac{1}{m}A^2E - \frac{\pi^2}{m}\frac{ED^2}{A^2} + \frac{2}{m}AE^2\right)\,.
\end{aligned} \tag{50}$$

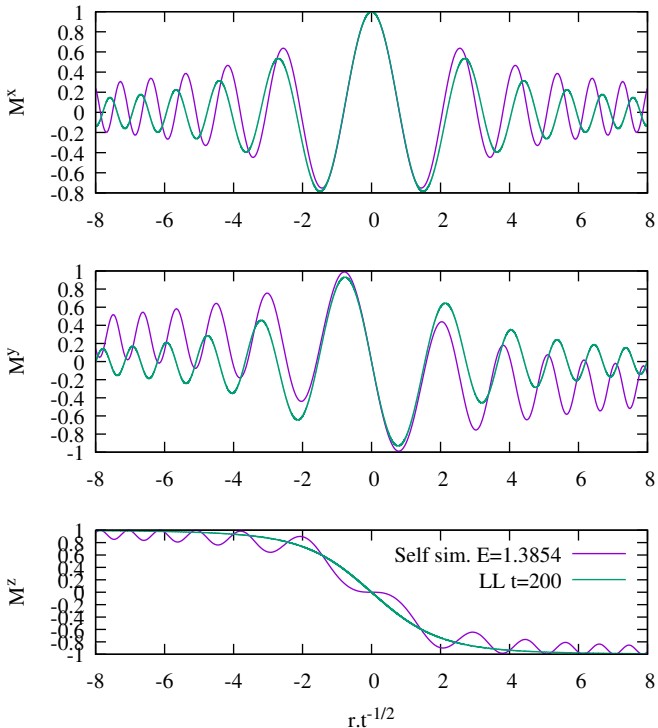

Figure 19: Components of the magnetization $\vec{M}(r,t)$ in the isotropic LL problem at $t = 200$ (green lines) compared with a self-similar solution (see text) parametrized by $E$ (purple lines). The value of $E$ is chosen so that both profiles have the same energy density at $r = 0$ (and $t = 200$). For $t = 200$ we have $\epsilon(r = 0, t) \cdot t \simeq 0.4798$ (as can be seen in the upper panel of Fig. 13), and $E = \sqrt{4\epsilon \cdot t} \simeq 1.3854$.

In the last expression one has written $B = \sqrt{AE}$. The Lagrange equations of motion corresponding to the Lagrangian $L$ show that $E$ is constant (this corresponds to the conservation of the norm $\int |\psi_{\text{var}}|^2 \mathrm{d}r = 2E/(gm)$) and that

$$6\dot{C} = -\frac{\pi^2}{2}\frac{\dot{D}}{A^2} - \frac{A^2}{m} - \frac{\pi^2}{m}\frac{D^2}{A^2} + \frac{4AE}{m}\,, \tag{51a}$$

$$\frac{\mathrm{d}}{\mathrm{d}t}\left(\frac{E}{A^2}\right) = \frac{4}{m}\frac{DE}{A^2}\,, \tag{51b}$$

$$0 = \frac{\pi^2\dot{D}E}{A^3} - \frac{2}{m}AE + \frac{2\pi^2}{m}\frac{D^2E}{A^3} + \frac{2}{m}E^2\,. \tag{51c}$$

A simple check is in order here: if one takes the initial condition $A(0) = A_0$, $E(0) = A_0$, $C(0) = 0$ and $D(0) = 0$ one indeed finds the soliton solution (48) with $C(t) = A_0^2 t/2m$.

In the general case, one will take $A(0) = A_0$, $C(0) = D(0) = 0$ and $E(0) = \alpha^2 A_0 \equiv E_0$, where $\alpha$ is a positive constant, so that one has initially

$$\psi_{\text{var}}(r, 0) = \alpha\,\psi_{\text{sol}}(r, 0)\,. \tag{52}$$

It is well known that for $\alpha$ an integer greater than one, the initial state (52) evolves into an oscillating bound state of $\alpha$ solitons [62]. The above variational ansatz cannot treat such cases. On the other hand, for $\alpha$ close to 1 we expect this approximation to correctly describe the nonlinear dynamics. In terms of the LL description, $\alpha$ parametrizes a family of planar

initial magnetization profiles

$$\vec{M}(r, t = 0) = \begin{pmatrix} \sin\theta(r) \\ 0 \\ \cos\theta(r) \end{pmatrix} \quad \text{with} \quad \theta(r) = 2\alpha \arccos(\tanh(-A_0 r)), \tag{53}$$

with an angle difference $\gamma = 2\alpha\pi$ (as expected from (44)).

Here we will push the approximation to $\alpha = \frac{1}{2}$, which is the case of interest since the initial condition of Eq. (45) corresponds to $A_0 = 2$ and $\alpha = 1/2$. Said differently, we consider a case where $\psi(t = 0)$ is "half" of a static soliton solution.

The next section explains how to solve the equations Eqs. (51). The comparison with the numerical results will be done in Sec. 7.7.2.

### 7.7.1 Solution of the variational dynamics

Factorizing out $E = E_0$ from Eq. (51c) and rewriting (51a) and (51c) one gets

$$-\frac{\pi^2}{2A^2}\left(\dot{D} + \frac{2D^2}{m}\right) + \frac{A}{m}(4E_0 - A) = 6\dot{C}, \tag{54a}$$

$$\frac{\pi^2}{A^3}\left(\dot{D} + \frac{2D^2}{m}\right) + \frac{2}{m}(E_0 - A) = 0, \tag{54b}$$

which shows that

$$6\dot{C} = \frac{A}{m}(5E_0 - 2A). \tag{55}$$

Eq. (51a) shows that

$$D = -m\dot{A}/(2A), \tag{56}$$

which, when inserted into (54b) yields

$$\frac{\mathrm{d}}{\mathrm{d}t}\left(-\frac{\dot{A}}{A^2}\right) + \frac{4A^2}{m^2\pi^2}(E_0 - A) = 0. \tag{57}$$

Defining the width $W = 1/A$ of the variational solution (49), this reads

$$\ddot{W} + \frac{4}{m^2\pi^2 W^2}\left(E_0 - \frac{1}{W}\right) = 0, \tag{58}$$

which admits the first integral

$$\frac{\dot{W}^2}{2} + \Gamma(W) = \Gamma_0, \tag{59}$$

$$\text{where} \qquad \Gamma(W) = \frac{2}{m^2\pi^2}\left(\frac{1}{W} - \alpha^2 A_0\right)^2, \tag{60}$$

$$\text{and} \qquad \Gamma_0 = \frac{2A_0^2}{m^2\pi^2}(\alpha^2 - 1)^2. \tag{61}$$

It is easy to see that if $\alpha < 1/\sqrt{2}$ one has $\Gamma_0 > \Gamma(+\infty)$ and the motion is not bounded. Such a situation is illustrated in Fig. 20. Since the case of interest is $\alpha = 1/2$ we are in a situation of this type.

The exact dynamics indeed shows two regimes, depending on the value of $\alpha$. For $\alpha \leq \frac{1}{2}$ the solution spreads monotonously, while for $\frac{3}{2} > \alpha > \frac{1}{2}$ the nonlinear structure oscillates around a one-soliton solution [62, 64]. The variational approach thus qualitatively captures

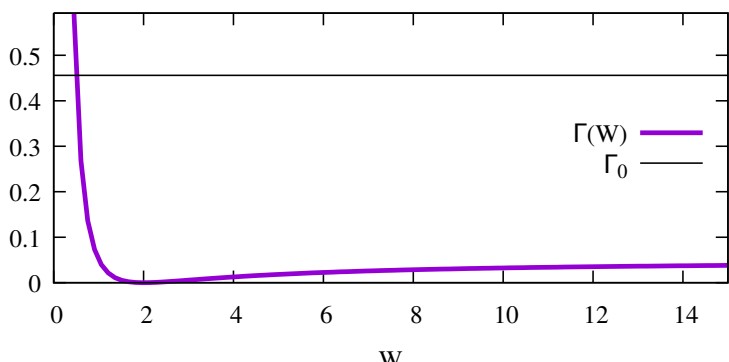

Figure 20: Effective potential $\Gamma$ for the width $W(t)$ [see text and Eq. (59)] for $m = 1$, $g = 1/4$, $A_0 = 2$ and $\alpha = 1/2$.

the two regimes, but not the exact value of the transition point. Still, for $\alpha = 1/2$, the present variational calculation correctly predicts an unbounded spreading.

To get the time evolution of $W$ one writes $\sqrt{2}\, dt = dW/\sqrt{\Gamma_0 - \Gamma(W)}$. In the case $\alpha = 1/2$ we are interested in, defining

$$w = W/W_0 = A_0 W, \tag{62}$$

one finds

$$
\begin{aligned}
\frac{\sqrt{2}}{m\pi W_0^2}\int_0^t dt &= \int_1^w \frac{w\, dw}{\sqrt{(w-1)(w+2)}} \\
&= \sqrt{w^2+w-2} - \frac{1}{2}\ln\left(1 + 2w + 2\sqrt{w^2+w-2}\right) + \frac{\ln 3}{2}\,. 
\end{aligned} \tag{63}
$$

This implies that, at large times, $W(t) \propto t$ with (additive) logarithmic corrections. Then (56) yields

$$D(t) = \frac{m}{2}\frac{\dot{w}}{w} = \frac{\sqrt{(w-1)(w+2)}}{\pi\sqrt{2}\, W_0^2 w^2}\,, \tag{64}$$

which implies that, at large times, $D \propto t^{-1}$ with logarithmic corrections. For solving the variational problem one just needs to invert Eq. (63) to get $w = w(t)$; then $A = A_0/w$, $D$ is given from (64) and from (55) one gets

$$\frac{dC}{dw} = \frac{\pi}{6\sqrt{2}}\frac{1}{\sqrt{(w-1)(w+2)}}\left(\frac{5}{4} - \frac{2}{w}\right)\,, \tag{65}$$

which makes it possible to express $C$ as a function of $w$. The expression is cumbersome, but at large $t$ it is of the form $\mathrm{cst} + \ln(t)$.

### 7.7.2 Comparison with numerics

As we explain now, the variational treatment discussed above reproduces a number of important features of the numerical solution. The modulus of the NLS wave-function is the curvature $\kappa$ in the LL problem: $|\psi| = \kappa$. In the variational approach we have $|\psi_{\mathrm{var}}| = \sqrt{\frac{E_0}{gmW(t)}}\{\cosh(r/W(t))\}^{-1}$. Since the LL energy density is $\epsilon = \frac{1}{8}\kappa^2$ we can express the energy density using $W(t)$: $\epsilon^{-1} \simeq 4W(t)\cosh(r/W(t))^2$ (we have used $E_0 = 1/2$). Since $W(t)$ grows linearly with time, the variational treatment correctly finds that $\epsilon \cdot t$ is a function of $r/t$.

For $r \ll t$, $|\psi_{\text{var}}|$ and the curvature $\kappa$ are approximately independent or $r$, and equal to $\kappa = \sqrt{\frac{E_0}{gmW(t)}}$, which goes to zero as $1/\sqrt{t}$. On the other hand, when the curvature is constant in space we may estimate the length scale over which the magnetization goes from the vicinity of $+\vec{e}_z$ to the vicinity of $-\vec{e}_z$ by $\kappa^{-1}$.[6] Here this length scale is proportional to $\sqrt{t}$. In other words the variational approximation predicts a diffusive expansion of the $M^z$ profile. As discussed previously this is correct up to logarithmic correction.

Going back to the energy density away from the origin, the variational approximation gives an exponentially decay at large $r/t$, which is in agreement with the exact result discussed in Sec. 7.4. Using Eq. (63) we find

$$W(t) \simeq \frac{2\sqrt{1-2\alpha^2}}{m\pi W_0} t , \quad \text{when } t \to +\infty . \tag{66}$$

For $\alpha = 1/2$, $W_0 = 1/2$ and $m = 1$ this gives $W(t) \sim t \cdot 2\sqrt{2}/\pi$. For large $r/t$ we get $\epsilon \cdot t \sim \exp(-c\, r/t)$ with $c = \pi/\sqrt{2} \simeq 2.22$, to be compared to $\simeq 1.6$ obtained from the fit in Fig. 15. This technically simple and physically intuitive approach thus reproduces the scaling of the energy profile at large distance. The modulus squared of the NLS wave-function, $|\psi_{\text{var}}|^2/8$, is compared with the energy density of the (isotropic) LL problem in Fig. 21. By construction the two curves coincide at $t = 0$ (not shown), and they remain close to each other at short times. The variational solution captures qualitatively the spread of the energy at long times. It does however not capture the logarithmic term observed in Fig. 16, hence the fact that the LL energy density at $r = 0$ decays more slowly with time ($\sim \ln(t)/t$) than the variational estimate ($\sim 1/t$).

It is also useful to consider the NLS current $J = \frac{1}{m}\text{Im}(\psi^*\psi_x)$. With the variational Ansatz it becomes

$$J_{\text{var}}(r, t) = \frac{2}{m} D\, r\, |\psi_{\text{var}}|^2 . \tag{67}$$

Eqs. (66), (56) and the above show that $D(t) = \frac{m}{2}\text{d}\ln(w)/\text{d}t \simeq m/(2t)$ at large $t$. These asymptotic estimates combined with (67) yield

$$J_{\text{var}}/|\psi_{\text{var}}|^2 \simeq r/t, \tag{68}$$

independently of the value of $\alpha$ and $W_0$. Since this ratio is also the space derivative $\theta_r$ of the complex argument of $\psi$, it gives, according to Eq. (39), the LL torsion. We thus recover the relation $\tau \simeq r/t$ obeyed by the solution of the LL problem (Fig. 18).

# 8 Conclusion

In this work we argued that the dynamics of the domain wall problem for a spin-$\frac{1}{2}$ XXZ chain close to the isotropic point behaves semiclassicaly at long time, and can asymptotically be described in the framework of an LL equation. This picture is supported by the systematic numerical comparison we made between the quantum and the classical problems, including an analysis of the in-plane component of the magnetization, the energy density or the energy current.

We identified the different scaling regimes for all these quantities, from the shortest length scale with $r \sim \mathcal{O}(1)$ to larger distances, from $r \sim \mathcal{O}(\sqrt{t})$ to $r \sim \mathcal{O}(t)$. The latter regime, that we call ballistic, is present in the LL problem as well as in the quantum model, in the easy-plane, isotropic and easy-axis cases. This regime, where the magnetization is almost pointing

---

[6]This amounts to approximate this part of the curve by a half circle.

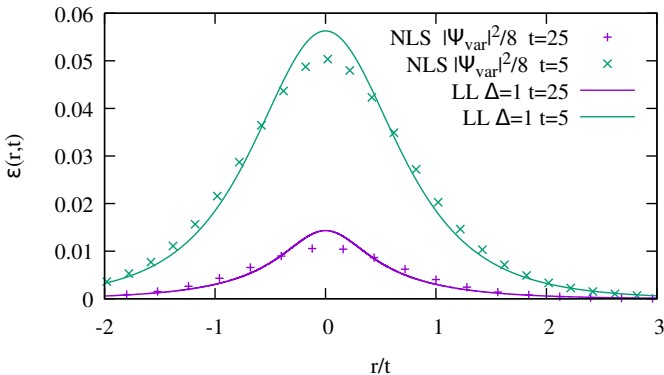

Figure 21: Energy density $\epsilon$ in the isotropic LL model compared with $|\psi_{\mathrm{var}}|^2/8$ from the variational solution of the NLS equation [Eq. (49)].

along the $z$ axis, can be captured by a perturbative expansion or, at $\Delta = 1$, by a simple variational calculation based on the isotropic LL to NLS mapping. The latter approximation predicts a diffusive expansion of the $z$ magnetization profile but misses the observed logarithmic correction (in the energy density, or in the width of the $z$ profile). Still at $\Delta = 1$, we showed that the LL magnetization in the diffusive regime can be described by a self-similar solution with an effective time-dependent curvature which incorporates the logarithmic enhancement. But, although a logarithmic divergence was identified in the inverse scattering data of the isotropic LL problem [27], it seems that a simple and/or intuitive explanation for the appearance of the logarithmic terms at $\Delta = 1$ is still missing, as well as an analytical expression for the asymptotic magnetization profile.

There are several possible future directions for developing the line of research exposed in the present work. The anisotropic LL problem has a great richness and can give rise to several other types of nonlinear phenomena, like dispersive shock waves or rarefaction wave, see for instance Ref. [40]. We could also mention some fascinating polygonal structures that can emerge in the isotropic case for particular initial conditions [65]. It would be interesting to investigate if some other initial conditions in the quantum spin chain could realize these phenomena.

Exploring the quantum corrections to the classical behavior in the quantum chain also seems a promising direction for future studies, and could take the form of some $1/S$ expansion. As for the entanglement entropy, while the heuristic argument given in Sec. 7.2 in favor of $S_{\mathrm{vN}} \sim \frac{1}{2} \ln t$ is simple and appealing, this prediction does not fully agree with the present numerics. The entanglement growth is certainly one of the interesting open questions in the domain wall problem at $\Delta = 1$.

## Acknowledgements

We thank V. Banica, A.M. Kamchatnov, K. Mallick, P. Krapivsky, J.-M. Stéphan and J. de Nardis for discussions about this problem. We are particularly grateful to K. Mallick and P. Krapivsky for sharing their unpublished results [28].

# A From isotropic LL to NLS

For completeness we provide here a detailed and self-contained derivation of the mapping from the isotropic LL equation to NLS [59–61]. The starting point is the isotropic LL equation:

$$2m\vec{M}_t = \vec{M} \wedge \vec{M}_{rr}, \tag{69}$$

with $\vec{M}^2(r,t) = 1$. The parameter $m$ was introduced to simplify the comparison with other possible conventions. To match the convention used in Sec. 7.5 one has to take $m = 1$.

To the 3-vector $\vec{M} = (M^1, M^2, M^3)$ we associate a $2 \times 2$ matrix

$$\mathbf{M} = M^1 \sigma_1 + M^2 \sigma_2 + M^3 \sigma_3, \tag{70}$$

where $\sigma_1$, $\sigma_2$ and $\sigma_3$ are the Pauli matrices. The magnetization $\vec{M}$ can be obtained by a rotation the $(0,0,1)$ vector. In terms of the associated 2 by 2 matrix, we can thus write

$$\mathbf{M}(r,t) = \mathfrak{g}^{-1} \sigma_3 \mathfrak{g}, \tag{71}$$

where $\mathfrak{g}(r,t) \in SU(2)$ implements the rotation. The space derivative of $\mathbf{M}$ is then

$$\mathbf{M}_r = \mathfrak{g}^{-1} [\sigma_3, \mathbb{L}] \mathfrak{g}, \tag{72}$$

where we have defined

$$\mathbb{L}(r,t) = \mathfrak{g}_r \mathfrak{g}^{-1}. \tag{73}$$

In a similar way the time derivative of the magnetization matrix reads

$$\mathbf{M}_t = \mathfrak{g}^{-1} [\sigma_3, \mathbb{M}] \mathfrak{g}, \tag{74}$$

where we have defined

$$\mathbb{M}(r,t) = \mathfrak{g}_t \mathfrak{g}^{-1}. \tag{75}$$

There is gauge freedom in the choice of the rotation since $\mathfrak{g}' = \mathfrak{g} \exp(i\Theta\sigma_3)$ with any angle $\Theta(r,t)$ would also be a valid choice in (71). In the following we chose a gauge such that the matrix $\mathbb{L}$ defined above has zeros on its diagonal:

$$\mathbb{L} = \begin{pmatrix} 0 & -\sqrt{gm}\,\bar{u}(r,t) \\ \sqrt{gm}\,u(r,t) & 0 \end{pmatrix}. \tag{76}$$

The parameter $g$ is arbitrary here and is simply introduced here to facilitate the comparison with other conventions found in the literature concerning the NLS equation. Note that the convention used in Sec. 7.5 amounts to take $g = 1/4$ and $m = 1$.

With the above gauge choice $\mathbb{L}$ anti-commutes with $\sigma_3$ and

$$[\sigma_3, \mathbb{L}] = -2\sqrt{gm}\left(u\sigma^+ + \bar{u}\sigma^-\right), \tag{77}$$

with $\sigma^+ = \begin{pmatrix} 0 & 0 \\ 1 & 0 \end{pmatrix}$ and $\sigma^+ = \begin{pmatrix} 0 & 1 \\ 0 & 0 \end{pmatrix}$. It is interesting to compute $\vec{M}_r^2$, since it proportional to the energy energy density in LL. From (72) we have $\vec{M}_r = -\det([\sigma_3, \mathbb{L}])$ and thus, from (77),

$$\vec{M}_r^2 = 4gm|u|^2. \tag{78}$$

As a check, if we use the convention of Sec. 7.6, i.e. $g = 1/4$ and $m = 1$, the equation above gives a curvature $\kappa = ||\vec{M}_r|| = |u|$. This is consistent with Eq. (39).

Combining (72) and (77) we get

$$\mathbf{M}_r = -2\sqrt{gm}\, \mathfrak{g}^{-1}\left(u\sigma^+ + \bar{u}\sigma^-\right)\mathfrak{g}. \tag{79}$$

Now we come back to the LL equation of motion. It can be written in terms of the magnetization current

$$\vec{J} = \frac{1}{2m}\vec{M} \wedge \vec{M}_r, \tag{80}$$

such that Eq. (69) becomes $\vec{M}_t = \vec{J}_r$. From (80) we get the matrix associated to $\vec{J}$ as a commutator:

$$\mathbf{J} = \frac{1}{4im}\left[\mathbf{M}, \mathbf{M}_r\right]. \tag{81}$$

Using Eqs. (71) and (79) this is expressed as

$$\mathbf{J} = \frac{-2\sqrt{gm}}{4im}\mathfrak{g}^{-1}\left[\sigma_3, u\sigma^+ + \bar{u}\sigma^-\right]\mathfrak{g} = \frac{1}{im}\mathfrak{g}^{-1}\mathbb{L}\mathfrak{g} = \frac{1}{im}\mathfrak{g}^{-1}\mathfrak{g}_r. \tag{82}$$

As for the current gradient, it reads

$$\mathbf{J}_r = \frac{1}{im}\mathfrak{g}^{-1}\mathbb{L}_r\mathfrak{g}. \tag{83}$$

Using Eq. (74) and the equation above, we can write the equation of motion $\mathbf{M}_t = \mathbf{J}_r$ as

$$[\sigma_3, \mathbb{M}] = \frac{1}{im}\mathbb{L}_r. \tag{84}$$

Using the explicit form of $\mathbb{L}$ (76), the equation above implies that $\mathbb{M}$ must have the following form:

$$\mathbb{M} = i\begin{pmatrix} \alpha & \frac{1}{2}\sqrt{\frac{g}{m}}\bar{u}_r \\ \frac{1}{2}\sqrt{\frac{g}{m}}u_r & -\alpha \end{pmatrix}, \tag{85}$$

where, so far, $\alpha$ is unknown. The fact that $\mathbb{M}$ must be traceless can be obtained from (73), since it implies that $\mathrm{Tr}\,\mathbb{M} = \frac{d}{dt}\det(\mathfrak{g})$. The latter derivative vanishes since $\det(\mathfrak{g}) = 1$. At this point we will consider two different expressions for $\mathfrak{g}_{rt}$. The first one can be obtained by taking $\mathfrak{g}_r$ from (73) and then taking the derivative with respect to time:

$$\mathfrak{g}_{rt} = \frac{d}{dt}\left(\mathbb{L}\mathfrak{g}\right) = \mathbb{L}_t\mathfrak{g} + \mathbb{L}\mathfrak{g}_t = \mathbb{L}_t\mathfrak{g} + \mathbb{L}\mathbb{M}\mathfrak{g}, \tag{86}$$

where, in the last expression, we have written $\mathfrak{g}_t$ using (75). Now we repeat the calculation of $\mathfrak{g}_{rt}$, doing the derivations in reverse order:

$$\mathfrak{g}_{rt} = \frac{d}{dr}\left(\mathbb{M}\mathfrak{g}\right) = \mathbb{M}_r\mathfrak{g} + \mathbb{M}\mathfrak{g}_r = \mathbb{M}_r\mathfrak{g} + \mathbb{M}\mathbb{L}\mathfrak{g}. \tag{87}$$

Comparing (86) and (87) we get the Lax equation:

$$\mathbb{L}_t - \mathbb{M}_r + [\mathbb{L}, \mathbb{M}] = 0. \tag{88}$$

The upper left element of the matrix equation above is

$$-i\alpha_r - \frac{ig}{2}(u\bar{u})_r = 0. \tag{89}$$

It means that $\alpha = -\frac{g}{2}|u|^2 + A(t)$ where $A$ is some integration constant, independent of $r$. We then write down the lower left element of the Lax equation:

$$\sqrt{gm}u_t - \frac{i}{2}\sqrt{\frac{g}{m}}u_{rr} + 2iu\alpha\sqrt{gm} = 0. \tag{90}$$

Replacing $\alpha$ by its expression found above we get:

$$\sqrt{gm}u_t - \frac{i}{2}\sqrt{\frac{g}{m}}u_{rr} + 2iu(-\frac{g}{2}|u|^2 + A(t))\sqrt{gm} = 0, \tag{91}$$

and after some rearrangement:

$$iu_t = -\frac{1}{2m}u_{rr} - gu\left(|u|^2 - \Lambda(t)\right), \tag{92}$$

with $\Lambda(t) = 2A(t)/g$. The equation above is the same NLS equation as (40) if we set $m = 1$ and $g = 1/4$.

To finish this short review of the mapping, it is interesting to interpret geometrically the (space- and time-dependent) $SO(3)$ rotation induced by $\mathfrak{g}$, which rotates the vector $(0,0,1)$ to the magnetization direction $\vec{M}$. To this end one defines the orthonormal basis $\{\vec{e}_1, \vec{e}_2, \vec{M}\}$ which is the image of the initial basis under the rotation. We then consider the $2 \times 2$ matrices associated to $\vec{e}_1$ and $\vec{e}_2$:

$$\mathbf{e}_1 = \mathfrak{g}^{-1}\sigma_1\mathfrak{g} \quad \text{and} \quad \mathbf{e}_2 = \mathfrak{g}^{-1}\sigma_2\mathfrak{g}. \tag{93}$$

Taking the space derivative of the equations above (as was done in (72)) we get:

$$\mathbf{e}_{1r} = \mathfrak{g}^{-1}[\sigma_1, \mathbb{L}]\mathfrak{g} \quad \text{and} \quad \mathbf{e}_{2r} = \mathfrak{g}^{-1}[\sigma_2, \mathbb{L}]\mathfrak{g}. \tag{94}$$

Using the explicit form (76) of $\mathbb{L}$ (Eq. 76) we find

$$\begin{aligned}
\mathbf{e}_{1r} &= 2\sqrt{gm}\,\mathrm{Re}(u)\sigma_3, \\
\mathbf{e}_{2r} &= 2\sqrt{gm}\,\mathrm{Im}(u)\sigma_3.
\end{aligned} \tag{95}$$

Going back to 3-vectors we can thus write:

$$\begin{pmatrix} \vec{e}_1 \\ \vec{e}_2 \\ \vec{M} \end{pmatrix}_r = 2\sqrt{gm} \begin{pmatrix} 0 & 0 & \mathrm{Re}(u) \\ 0 & 0 & \mathrm{Im}(u) \\ -\mathrm{Re}(u) & -\mathrm{Im}(u) & 0 \end{pmatrix} \begin{pmatrix} \vec{e}_1 \\ \vec{e}_2 \\ \vec{M} \end{pmatrix}, \tag{96}$$

where, to obtain the last line, we have used the fact that the $3 \times 3$ matrix above, describing the space derivative of the rotation which connects the initial basis to the local one, must be antisymmetric. We see in particular that the gauge choice made in (76) does not correspond to the local Serret-Frenet frame of the curve, but to its Bishop frame [66].

## B From NLS to isotropic LL

In this section we recall how, starting from a solution of the NLS equation, one can construct the associated solution of the isotropic LL equation. We start from a solution $u(r,t)$ of Eq. (40) and we wish to construct the associated LL magnetization $\vec{M}$. The starting point is Eq. (73), that we rewrite under the form

$$\mathfrak{g}_r = \begin{pmatrix} 0 & -\sqrt{gm}\,\bar{u} \\ \sqrt{gm}\,u & 0 \end{pmatrix}\mathfrak{g}. \tag{97}$$

We parametrize $\mathfrak{g}$ using two complex numbers $\mu$ and $\nu$ satisfying $|\mu|^2 + |\nu|^2 = 1$ as follows:

$$\mathfrak{g} = \begin{pmatrix} \mu & -\bar{\nu} \\ \nu & \bar{\mu} \end{pmatrix}. \tag{98}$$

Looking at the first column of Eq. (97), we get the following equations for $\mu$ and $\nu$ and their space derivatives:[7]

$$\mu_r = -\sqrt{gm}\,\bar{u}\,\nu \tag{99}$$
$$\nu_r = \sqrt{gm}\,u\,\mu. \tag{100}$$

After eliminating $\nu$ from these equations we obtain a linear differential equation for $\mu$:

$$\mu_{rr} - \mu_r \frac{\bar{u}_r}{\bar{u}} + gm|u|^2\mu = 0. \tag{101}$$

Note that the above derivation has the advantage of giving directly a *linear* differential equation for $\mu$. We are interested in situations where the magnetization points in the $x$ direction at $r = 0$. It is then convenient to adopt the following correspondence between the directions $1, 2, 3$ of Eq. (70) and the direction $x, y, z$: $x \to 3$, $-y \to 1$ and $-z \to 2$. In this way, $\mathfrak{g}(r = 0)$ is the identity and the initial condition for (101) is $\mu(r = 0) = 1$ (and $\nu(r = 0) = 0$). So, using Eqs. (70) and (71) we can relate $\mu$ and $\nu$ to the magnetization:

$$-M^1 = M^y = 2\,\text{Re}\,(\mu\nu) \tag{102a}$$
$$-M^2 = M^z = 2\,\text{Im}\,(\mu\nu) \tag{102b}$$
$$M^3 = M^x = |\mu|^2 - |\nu|^2 = 2|\mu|^2 - 1. \tag{102c}$$

## C  Magnetization for the self-similar solutions of the isotropic LL equation

We provide here a detailed solution of Eq. (101), making it possible to obtain some explicit expression for the self-similar solutions of the isotropic LL equation. The final result [Eqs. (107) and (109)] is expressed in terms of some confluent hypergeometric functions (Kummer function $_1F_1$). Some of these results have been found previously in Refs. [54, 58].

We start with the following filament function, parametrized by $E$:

$$u(r, t) = \frac{E}{2\sqrt{t}} \exp\left(\frac{ir^2}{4t}\right). \tag{103}$$

It is a solution of

$$iu_t = -u_{rr} - gu\left(|u|^2 - \frac{E^2}{4t}\right). \tag{104}$$

It corresponds to $m = \frac{1}{2}$ and $\Lambda(t) = \frac{E^2}{4t}$ in Eq. (92). We then set $g = 2$. Since $\bar{u}_r/\bar{u} = -ir/(2t)$, when we plug (103) into (101) we obtain [57]:

$$\mu_{rr} + \frac{ir}{2t}\mu_r + \frac{E^2}{4t}\mu = 0. \tag{105}$$

---

[7]Note that $(\mu, \nu)$ may be viewed a quantum state for a spin-$\frac{1}{2}$: $|\phi\rangle = \begin{pmatrix} \mu \\ \nu \end{pmatrix}$. If we interpret $r$ as a fictitious time, the equations above describe the motion of a spin in some "time"-dependent magnetic field $\vec{B}(r)$. Indeed we have $i\frac{d}{dr}|\phi\rangle = H|\phi\rangle$ with an Hamiltonian $H = B^1\sigma_1 + B^2\sigma_2 = \sqrt{gm}[\text{Re}(u)\sigma_2 - \text{Im}(u)\sigma_1]$, that is $\vec{B} = \sqrt{gm}(\text{Re}(u), -\text{Im}(u), 0)$.

We then replace the variable $r$ by $z = -ir^2/(4t)$. After a few manipulations we arrive at

$$\mu_{zz} + \mu_z \left( \frac{1}{2} - z \right) + \frac{iE^2}{4} \mu = 0. \tag{106}$$

With the initial condition $\mu(0) = 1$, the solution of this equation is the Kummer (confluent hypergeometric) function

$$\mu(r,t) = {}_1F_1 \left( -\frac{iE^2}{4}, \frac{1}{2}, -\frac{ir^2}{4t} \right). \tag{107}$$

We can, using Eq. (99), get the other matrix element of $\mathfrak{g}$, which satisfies $\nu(r=0,t) = 0$:

$$\nu(r,t) = -\frac{\mu_r}{\bar{u}} = \frac{2\sqrt{t}}{E} e^{\frac{ir^2}{4t}} \mu_r. \tag{108}$$

And since $\frac{d}{dz} {}_1F_1(p,q,z) = \frac{p}{q} {}_1F_1(p+1, q+1, z)$ we find [58]:

$$
\begin{aligned}
\nu(r,t) &= \frac{2\sqrt{t}}{E} \left( \frac{-iE^2}{2} \right) \left( \frac{-ir}{2t} \right) e^{\frac{ir^2}{4t}} {}_1F_1 \left( -\frac{iE^2}{4} + 1, \frac{3}{2}, -\frac{ir^2}{4t} \right) \\
&= -\frac{ar}{2\sqrt{t}} e^{\frac{ir^2}{4t}} {}_1F_1 \left( -\frac{iE^2}{4} + 1, \frac{3}{2}, -\frac{ir^2}{4t} \right).
\end{aligned} \tag{109}
$$

These expressions for $\mu$ and $\nu$ can be used to obtain the components of $\vec{M}$, using Eqs. (102). A comparison between such a self-similar solution and the solution of the LL domain wall problem with a smooth initial condition is shown in Fig. 19 and is discussed at the end of Sec. 7.5.

As a consistency check, one can verify that the formula above yields a curvature $||\vec{M}_r|| = \kappa = E/\sqrt{t}$. This is consistent with Eq. (103) and Eq. (78) with $g = 2$ and $m = 1/2$. One can also check that the torsion $\tau = \kappa^{-2} \vec{M} \cdot \left( \vec{M}_r \wedge \vec{M}_{rr} \right)$ is equal to $r/(2t)$. This implies that the time variable $t$ used in the calculation above is in fact $\tilde{t}$ in the notations of Sec. 7.5. It is thus straightforward to convert the present formula for the self-similar solution $\vec{M}(r,t)$ to the convention $m = 1$ by the replacement $t \to t/2$, as was done in Fig. 19.

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
