# Peer review of "Domain wall problem in the quantum XXZ chain and semiclassical behavior close to the isotropic point"

_SciPost Physics, doi:SciPost Phys. 7, 025 (2019)_

## Round 1 · Referee Report · Anonymous (Referee 1) · 2019-6-17

Strengths

1- Direct confirmation of the quantum-classical correspondence of spin transport in the anisotropic Heisenberg model. 2- Careful and detailed analysis with high-quality numerical data.

Weaknesses

1- Lack of integrability.

Report

This is an extensive study of the domain-wall dynamics in the Heisenberg quantum spin-1/2 chain with anisotropic interaction. There has been a number of works devoted to this problem in the past. The most recent study, ref. [27], already addressed the problem in the context of the classical Landau-Lifshitz field theory using integrability and other analytical tools, provided theoretical explanations for three distinct dynamical regimes of spin transport which strongly hint on a remarkable quantum-classical correspondence.

In the present work, the authors carry out a thorough quantitative analysis on this problem which further vindicates the conjectured quantum-classical correspondence. Specifically, they demonstrate that, on sufficiently large time-scales (which crucially depend on the anisotropy parameter), the asymptotic regions effectively behave as large ferromagnetic reservoirs which motivates a semi-classical description. By conjecturing that the intermediate (that is interface) region is simply governed by the classical Landau-Lifshitz equation of motion, they go on to convincingly demonstrate that the effective classical description becomes even quantitatively exact provided the interaction anisotropy simultaneously scales zero. Extensive numerical (tDMRG) simulations are supplemented by theoretical arguments based on various approximate models. In addition, the authors briefly address energy transport through the spreading of a localized energy-density profile, where they exploit the known mapping to the focusing NLS equation and solve it approximately using a variational Ansatz. They conclude that the variational approach is not sufficient to capture also the logarithmic deviations of the diffusive spreading.

In my opinion this is a well-rounded and nicely written paper, with a good mixture of analytic approaches and high-quality numerical simulations. Given however that model is exactly solvable, it is a pity that it does not offer any in-depth theoretical understanding using techniques of classical and quantum integrability. But in spite of that, I definitely do recommend this paper for publication after a small revision.

Comments:

There is a somewhat awkward language at certain places which I would like to address first:

i) At the beginning of Sec. 4, it is said " We give below an alternative -- and somewhat direct -- derivation of this solution."

So far I can see, the presented derivation is exactly the same as that presented previously in ref. [27]. Well, this is just the simplest Riemann-type hydrodynamic approximation where one neglects the dispersive term and solves the 'Riemann flow' subjected to suitable boundary conditions. This derivation is nothing deep really, merely a short exercise. I do not think it is legitimate to proclaim this derivation as "alternative" or "more direct"...

ii) At the beginning of Sec. 6, we can read " As for the LL problem, the z magnetization problem was observed to freeze at long times [27, 28], and to be described by a soliton solution of the easy-axis LL equation."

One should notice here that ref. [27] obtained the exact spectral data for the domain-wall initial profile. In the easy-axis regime, the spectrum involves a static kink configuration (in general there are additional radiative modes and/or breathers). The absence of spin transport is thus attributed exclusively to the stability of a localized kink, which an exact (and rigorous) statement. Saying "was observed to freeze" may read as an understatement.

Other remarks and questions:

1) I think it would be beneficial to clarify a bit better how much can one strictly infer from the energy conservation argument. Just by knowing that asymptotic regions will remain ferromagnetic does not automatically ensure that the interface region centred at the origin will be free of genuine quantum effects (and therefore effectively governed by the classical ferromagnetic field theory), does it? In my reading, the argument here relies on the empirical evidence (based on e.g. [26] or other simulations) that the magnetization gradient vanishes as $t \to \infty$ on the characteristic length R(t). Naively, one would then expect the semi-classical description should work outside of the dynamical interface [-R(t),R(t)]. But then what guarantees you that long-wavelength semi-classical theory w.r.t. ferromagnetic vacuum remains applicable in the vicinity of the origin (even at $\Delta = 1$ to begin with)?

2) In order to observe the quantum-classical correspondence at the quantitative level one has to probe the dynamics in the weakly anisotropic regime. Then I am curious whether one could also detect classical breather modes (which "live" in the strongly anisotropic regime of the Landau-Lifshitz magnet) in the non-equilibrium dynamics of the quantum spin chain?

3) If the authors could give some quantitative estimates on the "quantum corrections" (when moving away from the $\Delta \to 1^\pm$), that would be very nice.

Other remarks:

  • The presentation of the left plot from Fig. 9 can be probably be improved. In particular, it is hard to distinguish the blue and yellow curves by naked eye. Moreover, I have a question here: how does one determine the fitting window? Is there a way to estimate the relevant transient time-scale after which the asymptotic law become effectively valid I can imagine that by some ad-hoc manipulations (e.g. sliding the window to larger times), the green fit for the diffusive curve will presumably look much better. Perhaps some clarifications along these line can be helpful.

Minor remarks:

  • Bad spelling: "insure" should be "ensure" (end of page 23).
  • Just a suggestion which may slightly improve readability: one could try use explicit symbols to the terms of the sort $1/4 - S \cdot S$ and $1/4 - S^{z} \cdot S^{z}$, e.g. using $H_{iso}$ and $V$.

---

## Round 1 · Referee Report · Anonymous (Referee 2) · 2019-6-30

Strengths

1- Semiclassical description of the out of equilibrium Heisenberg chain close to the isotropic point in terms of the classical Landau Lifshitz equation. 2- State of the art numerics on a difficult problem.

Weaknesses

1- Some overlap with Reference [27].

Report

This paper presents a study of the emerging classical behavior following a inhomogeneous quench from a domain wall state in the (integrable) spin-1/2 anisotropic Heisenberg chain. The particular setup has been studied in the past, and can be seen as a quantum version of the famous Riemann problem in classical hydrodynamics.

The aim is to better understand the vicinity of the isotropic point, for which it is very difficult to use standard quantum integrability tools such as the Thermodynamic Bethe Ansatz. The main claim is a quantitative description in terms of the integrable Landau Lifschitz (LL) integrable PDE, at least for local observables. This claim is checked numerically in considerable detail. A nice and simple heuristic argument, based on energy conservation, is also provided to justify the emergence of the classical LL equation.

The paper is well written and interesting. It also raises a number of interesting questions regarding the connection between out of equilibrium quantum dynamics and integrable PDEs. I recommend publication, provided the following minor issues are addressed.

1) A tanh-sinh profile similar to the initial data for the LL can also be implemented in the spin chain. Did the authors look whether the results presented here remain valid for such initial states?

2) Regarding the corrections to the linear behavior discussed at the end of page 10. It is not clear to me, because of this, how the agreement between LL and XXZ can be seen as quantitative in a precise sense. Also, it appears that the agreement is not perfect at the isotropic point either. While I understand that the match between the various models is qualitatively remarkable, it is not clear how this may be turned into exact statements, such as the one for the density profile stemming from generalized hydrodynamics.

3) Regarding the discussion of the entanglement entropy in the isotropic case. Does the $\log(R(t))$ argument still underestimates the entanglement entropy upon inclusion of multiplicative logarithm corrections to R(t)?

4) Page 13, last paragraph. 'exponential decay in $r/t$...This exponential decay is however cut for $r/t=1$'. I do not understand this sentence. Wouldn't it be more correct to say that there is no exponential decay in the quantum case, since $e^{-1}=cst$?

5) Page 21, caption to figure 11. '$t=185$' contradicts the claim '$t=210$' at the end of page 19. Also, inconsistent notations $\tau$ vs $dt$.

6) Section 7.3. It is not clear to me how the LL and XXZ chain calculations should match, since the assumption of ferromagnetic correlations breaks down at very large distances. Usually when dealing with hydrodynamics descriptions of quantum systems, it is necessary to 're-quantize' the classical solution in order to access long-range correlations. This is what is done in Luttinger liquid theory, for instance. Could the authors comment on that?

7) Section 7.6, last paragraph. Perhaps it would be better to state more clearly that the divergences are precisely the logarithmic divergences that make up for the logarithmically enhanced diffusion discussed before.

Typos:

Page 2, 'form' -> 'from'. Page 4, line 12. 'and in characterize'. Page 4, the last sentence of the introduction starts with 'And'. Page 10, last line: '$r/r$' ->'$r/t$'. Page 18, second paragraph in 7.1. 'As can be seen in in' -> 'As can be seen in' Page 26, line before equation (39). 'the following the filament' ->'the following filament'. Also, the filament function is '$\psi(x,t)$' not $u(x,t)$, so the sentence reads awkwardly. Page 26, two equations are not numbered.

Requested changes

See report.

---

## Round 1 · Referee Report · Anonymous (Referee 3) · 2019-7-3

Strengths

  • Timely topic and interesting results
  • Careful and exhaustive analysis

Weaknesses

  • Interpretation of entropy growth may be too naive

Report

The authors study the time evolution of a domain wall
in the XXZ chain and propose an effective description
of the dynamics via the anisotropic Landau-Lifshitz
(LL) equation. The results of DMRG calculations on the
XXZ are compared to the simulations of the LL dynamics
for various observables. If the LL anisotropy parameter
is fixed in an appropriate way, the results show a
quite good qualitative agreement. On the top of that
it is found that, in the isotropic Heisenberg limit,
the agreement becomes even quantitative, suggesting
that the quantum dynamics of the domain wall becomes
effectively classical.

The manuscript presents a very detailed and exhaustive
analysis of the problem at hand, with a particular
focus on the $\Delta=1$ case. In particular, the authors
identify two different scaling regimes of the dynamics,
namely a diffusive one (with logarithmic corrections)
as well as a ballistic tail regime. The numerical results
are corroborated by perturbative analytical calculations
with a good agreement.

In my opinion this is a well written manuscript that
treats a problem of current interest, and thus well
deserves to be published. I have only one point of
criticism related to the calculation of the entropy.

Requested changes

The main issue:

I have serious doubts about the correctness of the
heuristic argument presented by the authors to interpret
the entanglement entropy growth. Indeed, they essentially
invoke only the U(1) symmetry, which is present for
arbitrary values of $\Delta$. However, the authors
themselves make a footnote that the argument does not
apply away from $\Delta=1$. But why?
In particular, the same argumentation would give
$R(t) \sim t$ and thus $S(t) \sim \ln t$ for $\Delta < 1$
which is not the correct prefactor.

I believe, that this argument is just too naive to be able
to interpret the entanglement. In fact, having a look at
Fig. 11, one even observes that the ansatz log(t)/2 gives
a rather poor description of the numerical data.

The smaller corrections:

(1) In Fig. 4 the labels (1) and (2) should be shifted
in order not to overlap with the x axis. There is also
an extra parenthesis in the y axis label of (2).

(2) In Fig. 7, there is a typo in the legend,
$\Omega^x$ should read $\Omega^z$.

(3) In Fig. 8 is $I(t)$ the local (at $r=0$) or the global
current plotted? How comes that it is nonzero at $t=0$?

(4) Before Eq. (39): “filament function” $\psi(r,t)$
I guess this should read $u(r,t)$.

(5) Typo later —> latter (appears many times in text)

---

## Round 2 · Referee Report · Anonymous · 2019-7-31

Report
The authors have successfully responded to the criticism and introduced the necessary charges in the manuscript. The paper is now suitable for publication.

---

## Round 2 · Referee Report · Anonymous · 2019-7-31

Report
The authors have convincingly addressed all the issues in my previous report, the manuscript could be accepted as it is.

---

## Round 2 · Referee Report · Anonymous · 2019-8-20

Report
All my previous criticism has been addressed, the paper can be published now.

---

## Round 2 · Author Response

=============================================================
Reply to "Anonymous Report 1 on 2019-6-17 Invited Report"
=============================================================
> i) At the beginning of Sec. 4, it is said " We give below an
> alternative -- and somewhat direct -- derivation of this solution."
>
> So far I can see, the presented derivation is *exactly* the same as
> that presented previously in ref. [27]. Well, this is just the
> simplest Riemann-type hydrodynamic approximation where one neglects
> the dispersive term and solves the 'Riemann flow' subjected to
> suitable boundary conditions. This derivation is nothing deep really,
> merely a short exercise. I do not think it is legitimate to proclaim
> this derivation as "alternative" or "more direct"...
The referee is right, the derivation of this result is indeed quite
simple. Apart from the fact that we use polar coordinates and that we
do not make explicit use of the Riemann invariant $r_\pm$, the
derivation we present is essentially the same as that given by Gamayun
et al. The formulation has been changed in the revised version.
> ii) At the beginning of Sec. 6, we can read " As for the LL problem,
> the z magnetization problem was observed to freeze at long times [27,
> 28], and to be described by a soliton solution of the easy-axis LL
> equation." One should notice here that ref. [27] obtained the exact
> spectral data for the domain-wall initial profile. In the easy-axis
> regime, the spectrum involves a static kink configuration (in general
> there are additional radiative modes and/or breathers). The absence
> of spin transport is thus attributed exclusively to the stability of
> a localized kink, which an exact (and rigorous) statement. Saying
> "was observed to freeze" may read as an understatement.
We agree with the referee (and changed the text accordingly).
> 1) I think it would be beneficial to clarify a bit better how much
> can one strictly infer from the energy conservation argument. Just
> by knowing that asymptotic regions will remain ferromagnetic does
> not automatically ensure that the interface region centered at the
> origin will be free of genuine quantum effects (and therefore
> effectively governed by the classical ferromagnetic field theory),
> does it? In my reading, the argument here relies on the empirical
> evidence (based on e.g. [26] or other simulations) that the
> magnetization gradient vanishes as t→∞ on the characteristic length
> R(t). Naively, one would then expect the semi-classical description
> should work outside of the dynamical interface [-R(t),R(t)]. But
> then what guarantees you that long-wavelength semi-classical theory
> w.r.t. ferromagnetic vacuum remains applicable in the vicinity of
> the origin (even at Δ=1 to begin with)?
Yes, knowing that that asymptotic regions will remain ferromagnetic
does not automatically ensure that the interface region centered at the
origin will be free of genuine quantum effects. But the energy
conservation argument we put forward is different. We start from the
assumption that $R(t)\to\infty$ (which is the case for $|\Delta|\leq
1$). Then this implies that the total energy -- which is
$\mathcal{O}(1)$ gets "diluted" across the system. As a consequence,
for any fixed position $r$ (including the center at $r=0$), the energy
density $\epsilon(r,t)$ goes to zero when $t\to\infty$. Then, if
$\Delta=1$ the vanishing energy density implies a ferromagnetic state
(at least locally).
> 2) In order to observe the quantum-classical correspondence at the
quantitative level one has to probe the dynamics in the weakly
anisotropic regime. Then I am curious whether one could also detect
classical breather modes (which "live" in the strongly anisotropic
regime of the Landau-Lifshitz magnet) in the non-equilibrium
dynamics of the quantum spin chain?
If one goes deeply in the easy-axis regime of the quantum spin chain,
the magnetization profile hardly evolves before it rapidly freezes,
and the width of the asymptotic profile is close two the initial one,
that is one lattice spacing. So, the short answer is no. As far as we
can judge from our numerics, one does not detect the quantum analog of
the classical breather modes.
> 3) If the authors could give some quantitative estimates on the
"quantum corrections" (when moving away from the Δ→1±), that would
be very nice.
The referee is certainly right, it would be nice to have some better
understanding of these quantum corrections, and we are planning to
investigate this further in the future. At the moment we must however admit that we
do not yet have any result, except for the observation that some
quantities do not match perfectly between the Landau-Lifshitz problem
and the quantum spin chain (see for instance Fig. 10, 12 and 13).
> The presentation of the left plot from Fig. 9 can be probably be
improved. In particular, it is hard to distinguish the blue and
yellow curves by naked eye. Moreover, I have a question here: how
does one determine the fitting window? Is there a way to estimate
the relevant transient time-scale after which the asymptotic law
become effectively valid I can imagine that by some ad-hoc
manipulations (e.g. sliding the window to larger times), the green
fit for the diffusive curve will presumably look much better.
Perhaps some clarifications along these line can be helpful.
We have improved the readability of the left panel of Fig. 9 (by a
better choice of colors, and by plotting the Landau-Lifshitz data
using crosses instead of a full line).
The fitting window was chosen here simply to illustrate the fact that
some fitting functions [(b), (c) and (d)] continue to agree with the
data beyond (and before) the window, while the simple "diffusive"
function ((a), without logarithmic correction) only agrees inside the
window. The window is otherwise somewhat arbitrary. Sliding the
fitting window would indeed improve the agreement of the fit (a) up to
time $\sim 800$, but it would deteriorate at shorter times, and the
fitting function would presumably still strongly depart from the data
at longer times. It is not obvious if one can determine a finite
time-scale beyond which the asymptotic law become effectively
valid. Indeed, if we consider the fitting function (d) [Fig. 9], it
appears that the 1/t term becomes smaller than the constant term if
$t\gtrsim 1.8$. But for the constant term to become smaller than the
(dominant) logarithm, one needs to reach $t\sim 9000$.
=======================================================================
Reply to "Anonymous Report 2 on 2019-6-30 Invited Report"
=======================================================================
> 1) A tanh-sinh profile similar to the initial data for the LL can
also be implemented in the spin chain. Did the authors look whether
the results presented here remain valid for such initial states?
No, we have only considered the "sharp" domain wall state, where
$S^z=\pm1/2$. One attracting feature of this state is the absence of
any "parameter" and its invariance under rotations about the $z$ axis,
but it is clear that several other states seem worth investigating,
including those proposed by the referee.
> 2) Regarding the corrections to the linear behavior discussed at the
end of page 10. It is not clear to me, because of this, how the
agreement between LL and XXZ can be seen as quantitative in a
precise sense. Also, it appears that the agreement is not perfect
at the isotropic point either. While I understand that the match
between the various models is qualitatively remarkable, it is not
clear how this may be turned into exact statements, such as the one
for the density profile stemming from generalized hydrodynamics.
From our numerics (Fig. 3), the quantitative agreement between LL and
XXZ appear to be very good for $\Delta=0.8,0.9$ and 0.95, if we omit
the region corresponding to the tip of the front, where the profile
has not yet converged to its asymptotic form (for $\Delta=0.99$ the
result is expected to be good too, but for the accessible times the
profiles are still far from "converged"). In fact for $\Delta<1$ we
know from GHD and from the LL equation that both (quantum and
classical) profiles will become linear at long times, with two slopes
which become identical when $\Delta\to 1^-$. So, in the weakly
easy-plane regime there is no doubt that the $z$ magnetization
profiles become identical. The agreement for the in-plane component is
also excellent (right panel of Fig. 2). However, as noted by the
referee, it is at the isotropic point that the agreement is only
semi-quantitative (Fig. 10 for instance).
> 3) Regarding the discussion of the entanglement entropy in the
isotropic case. Does the log(R(t)) argument still underestimates the
entanglement entropy upon inclusion of multiplicative logarithm
corrections to R(t)?
Yes, even with the the multiplicative logarithm, log(R(t))
underestimates the entanglement entropy.
> 4) Page 13, last paragraph. 'exponential decay in r/t... This
exponential decay is however cut for r/t=1'. I do not understand
this sentence. Wouldn't it be more correct to say that there is no
exponential decay in the quantum case, since e−1=cst ?
Yes, the referee is right, our formulation was incorrect.
> 6) Section 7.3. It is not clear to me how the LL and XXZ chain
calculations should match, since the assumption of ferromagnetic
correlations breaks down at very large distances. Usually when
dealing with hydrodynamics descriptions of quantum systems, it is
necessary to 're-quantize' the classical solution in order to access
long-range correlations. This is what is done in Luttinger liquid
theory, for instance. Could the authors comment on that?
The referee is right, at a given time we should not expect the quantum
correlations to match the LL result beyond a certain distance. Even
if the spins have locally some ferromagnetic correlations, at any
finite time these correlations are not perfectly ferromagnetic.
Nevertheless, we have some indication that, at a fixed distance $r$,
the agreement between the XXZ correlations and the LL magnetization
improves with time. This can for instance be seen in the right panel
of Fig. 2, where the XXZ data at time $t=100$ appear to be closer to
the LL result than the XXZ data at $t=70$.
=============================================================
Anonymous Report 3 on 2019-7-3 Invited Report
=============================================================
> The main issue: I have serious doubts about the correctness of the
> heuristic argument presented by the authors to interpret the
> entanglement entropy growth. Indeed, they essentially invoke only
> the U(1) symmetry, which is present for arbitrary values of
> Δ. However, the authors themselves make a footnote that the argument
> does not apply away from Δ=1. But why? In particular, the same
> argumentation would give R(t)∼t and thus S(t)∼ln(t) for Δ<1 which is
> not the correct prefactor.
The first version of the manuscript was probably not clear enough on
that point, and we thank the referee for raising this issue.
In fact our argument does not apply away from Δ=1, because it does not
only rely on the U(1) symmetry. It is also necessary that the spins
acquire perfect ferromagnetic correlations, and this is only the case
at fixed position when $t\to \infy$ and Δ=1 (cf. energy conservation
argument). If we instead consider Δ slightly smaller than 1, the
nearest neighbor correlations will not converge everywhere to that of a
triplet. There will be small deviations from $\langle \vec S_r\cdot
\vec S_{r+} \rangle=1/4$. Although these small deviations may only
weakly affect the local magnetization of the energy density, it can
change completely the entanglement entropy. This can be illustrated,
for instance, by considering a simpler case. Consider the
ground-state of the xx chain, with a magnetization per site $m$ close
(but not equal) to $-1/2$. The spin-spin correlations can be made
arbitrarily close to that of a ferromagnet by taking $m$ to $-1/2$.
Nevertheless the quantum state is a (low but finite density) Fermi
sea. In such a case the entropy of a segment of length $l$ will scale
as $\sim \log(l)/3$. This is clearly very different from the $m=-1/2$
case (which is a product state).
> I believe, that this argument is just too naive to be able to
> interpret the entanglement. In fact, having a look at Fig. 11, one
> even observes that the ansatz log(t)/2 gives a rather poor
> description of the numerical data.
The referee is right, in the sense that log(t)/2 is only the
"classical" contribution to the entanglement entropy, corresponding to
a symmetrized classical state. And we also agree on the fact that the
numerics indicate that the actual entropy is larger than this
classical contribution (this has been made more clear in the new
version, including, in particular, in the caption of Fig. 11).
> (3) In Fig. 8 is I(t) the local (at r=0) or the global current
plotted? How comes that it is nonzero at t=0?
The current is indeed zero at t=0, as it should. But the initial
short-time increase of $I$ is quite abrupt at the scale of these
plots, and it was no therefore visible. It is corrected now.

---

## Round 2 · List of Changes

=============
From "Report 1"
=============
> I do not think it is legitimate to proclaim
this derivation as "alternative" or "more direct"...
* The formulation has been changed in the revised version.
* We have improved the readability of the left panel of Fig. 9 (by a
better choice of colors, and by plotting the Landau-Lifshitz data
using crosses instead of a full line).
> Minor remarks: Bad spelling: "insure" should be "ensure" (end of
page 23). Just a suggestion which may slightly improve readability:
one could try use explicit symbols to the terms of the sort 1/4−S⋅S
and 1/4−Sz⋅Sz, e.g. using Hiso and V
* Done.
> (...) Saying "was observed to freeze" may read as an understatement.
* We agree with the referee (and changed the text accordingly).
=============
From "Report 2"
=============
> (...) Wouldn't it be more correct to say that there is no
exponential decay in the quantum case, since e−1=cst ?
* Corrected.
> 5) Page 21, caption to figure 11. 't=185' contradicts the claim
't=210' at the end of page 19. Also, inconsistent notations τ vs dt.
* Corrected (we updated this plot, now the entropy data are available up
to t=400)
> 7) Section 7.6, last paragraph. Perhaps it would be better to state
more clearly that the divergences are precisely the logarithmic
divergences that make up for the logarithmically enhanced diffusion
discussed before.
* Done.
> Typos:
> Page 2, 'form' -> 'from'.
> Page 4, line 12. 'and in characterize'.
> Page 4, the last sentence of the introduction starts with 'And'.
> Page 10, last line: 'r/r' ->'r/t'.
> Page 18, second paragraph in 7.1. 'As can be seen in in' -> 'As can be seen in'
> Page 26, line before equation (39). 'the following the filament' ->'the following filament'.
> Also, the filament function is 'ψ(x,t)' not u(x,t), so the sentence reads awkwardly.
> Page 26, two equations are not numbered.
Corrected, thanks.
============
From Report 3
============
* We have improved the discussion on the entropy, at the end of Sec. 7.2.
We have explained why the semi-classical entropy argument does not
apply away from Δ=1.
> The smaller corrections:
>
> (1) In Fig. 4 the labels (1) and (2) should
> be shifted in order not to overlap with the x axis. There is also an
> extra parenthesis in the y axis label of (2).
> (2) In Fig. 7, there is a typo in the legend, Ωx should read Ωz.
* Done
* Improved Fig. 8, to make visible the fact that the current vanishes at t=0.
> (4) Before Eq. (39): “filament function” ψ(r,t) I guess this should
read u(r,t).
> (5) Typo later —> latter (appears many times in text)
Corrected

---

## Editorial Decision

published